# A multicenter longitudinal study of cholinergic subgroups in Parkinson disease

Nicolaas I. Bohnen [1,2,3,4,5,6,8] ✉, Stiven Roytman [1,2,4,8], Sygrid van der Zee[7], Giulia Carli[2,4,5], Fotini Michalakis [1,2,3,4], Austin Luker [1,2,3,4], Sofie Slingerland[7], Kirk A. Frey[1,2,5], Peter J. H. Scott[1], Robert A. Koeppe[1,2], Teus van Laar[7], Roger L. Albin[2,3,5] & Prabesh Kanel [1,2,3,4]

Parkinson disease (PD) is a heterogeneous syndrome. There is a need for biology-driven subtyping to inform specific therapeutic strategies. In a two-center study with de novo and established PD cohorts, we use vesicular acetylcholine transporter ligand [18F]FEOBV brain PET to assess cholinergic systems changes in early to moderate PD. Principal component analysis (PCA) is applied to data from 245 PD subjects to define cholinergic subgroups at baseline. Three PD subgroups are identified: hypercholinergic (regional upregulation; 29%), mixed (regional upregulation and regional deficits; 40.8%) and hypocholinergic (regional deficits only; 30.2%). Evidence of upregulation is observed in the subcortical-anterior cortical regions, whereas cholinergic downregulation is found in posterior cortical regions. Cholinergic upregulation and downregulation exhibit distinct associations with clinical symptoms. Longitudinal analysis (2-3 year interval) in 128 PD subjects reveals differential progressions by subgroup. This subtyping approach expands understanding of cholinergic progression in PD and may inform identification of new therapeutic targets.

Parkinson disease (PD) is a heterogeneous neurodegenerative syndrome with the defining feature of relatively early degeneration of nigrostriatal dopaminergic projections[1]. PD pathophysiology is multifactorial, reflecting dysfunction and degeneration of multiple central and peripheral nervous systems components[2]. Current subtyping approaches are mainly based on clinical features that inadequately reflect disease heterogeneity and natural history[3]. There is a need for objective, biologically informed subtyping that meaningfully reflects clinical heterogeneity, varying natural history, and informs development of more specific therapies[4]. Recent subtyping proposals focus on biomarkers correlating with a-synuclein deposition[5]. These proposals are controversial as not all forms of PD exhibit a-synuclein deposition

and this categorization may not reflect clinical and natural history heterogeneity[6,7]. Alternative, potentially complementary, biologically driven subtyping approaches could be based on relatively differential involvement of nervous system components[8-10]. More severe cholinergic denervation is implicated in a proposed 'malignant' subtype of PD[11-13], characterized by more severe dopaminergic medication-refractory postural instability and gait difficulties and cognitive decline/dementia[11,14]. Therefore, cholinergic system changes may uniquely predict clinically relevant disease milestones in PD.

[18F]fluoroethoxybenzovesamicol ([18F]FEOBV) is a selective positron emission tomography (PET) ligand for the vesicular acetylcholine transporter (VAChT), uniquely expressed by cholinergic terminals[15-17].

[1]Department of Radiology, University of Michigan, Ann Arbor, MI, USA. [2]Morris K. Udall Center of Excellence for Parkinson's Disease Research, University of Michigan, Ann Arbor, MI, USA. [3]Parkinson's Foundation Research Center of Excellence, University of Michigan, Ann Arbor, MI, USA. [4]Functional Neuroimaging, Cognitive, and Mobility Laboratory, Department of Radiology, University of Michigan, Ann Arbor, MI, USA. [5]Department of Neurology, University of Michigan, Ann Arbor, MI, USA. [6]Neurology Service and GRECC, VA Ann Arbor Healthcare System, Ann Arbor, MI, USA. [7]University Medical Center Groningen, Department of Neurology, University of Groningen, Groningen, the Netherlands. [8]These authors contributed equally: Nicolaas I. Bohnen, Stiven Roytman. ✉e-mail: nbohnen@umich.edu

[18F]FEOBV PET has excellent anatomic resolution and regional [18F]FEOBV binding likely reflects both cholinergic terminal integrity and cholinergic neuron activity[18,19]. The gene encoding VAChT, *SLC18A3*, is embedded in the first intron of the gene encoding choline O-acetyltransferase (ChAT; *CHAT*), the biosynthetic enzyme for acetylcholine (ACh)[20]. VAChT and ChAT expressions are subject to coordinated regulation[21]. Regional decreases in [18F]FEOBV binding are most parsimoniously interpreted as cholinergic terminal degeneration, though down-regulation of VAChT expression is plausible. Regional [18F]FEOBV binding above normal levels likely reflects a joint increase in VAChT and ChAT expression, possibly related to increased cholinergic neurotransmission.

Prior studies showed initial cholinergic denervation in posterior (parieto-occipital) cortices but potential up-regulation of cholinergic neurotransmission in other regions, especially in prodromal or earlier stage disease[10,14,22]. We previously reported bidirectional changes with evidence of posterior cortical cholinergic terminal losses but higher than normal [18F]FEOBV binding of both cortical and subcortical regions in de novo PD subjects with preserved cognition[22]. Increased regional [18F]FEOBV binding was absent in PD subjects with cognitive impairments[22]. Analogous results were reported by Denis-Legault et al. in a study of hippocampal [18F]FEOBV binding in PD subjects with and without cognitive dysfunction[23].

The purpose of the present multi-site, longitudinal, observational study was to explore a biology-based subgrouping by defining cholinergic brain system subgroups in PD using [18F]FEOBV PET. To develop a holistic characterization of the cholinergic system in PD, we apply a multi-scale analysis strategy which utilizes the normative deviation system-level cholinergic covariance networks to define the subgroups, voxel-level statistical parametric mapping (SPM) comparisons to identify more granular regional expression patterns, and whole-brain level summary statistics of cholinergic deviation spatial extent to capture the global trends of cholinergic system progression in PD. Furthermore, we leverage this multi-scale characterization of the cholinergic system in PD to shed greater light on its contributions to the manifestation and evolution of motor and non-motor symptoms, both cross-sectionally and longitudinally. We hypothesize that our biology-based subgrouping and multi-scale analysis strategy will capture the early cholinergic system upregulation (hyper-cholinergic) and late cholinergic system downregulation (hypo-cholinergic) processes reported on in previous works, and that the interplay between these two mechanisms will serve as a key predictor of clinical symptom severity and longitudinal trajectory.

## Results
### Subhead 1: demographics and clinical information
Demographic and clinical information for the combined and single center PD groups at baseline is presented in Table 1. In the baseline visit sample, Michigan and Groningen groups did not differ significantly on age or sex. The Groningen cohort had poorer cognition, greater impact of motor symptoms on activities of daily living, lower overall severity of motor symptoms, lower postural instability and gait difficulty (PIGD) symptom severity, earlier motor disease stage, and lower disease duration from symptom onset than the Michigan cohort. Both PD cohorts were matched with a pooled sample of control participants on age, had a greater proportion of males than observed among controls, and had poorer cognition than controls (see supplementary materials section 1).

Demographic and clinical information for the combined and single center PD groups at follow-up visit is presented in supplementary materials section 2, Table S1. In the follow-up sample ($N = 128$), Michigan ($N = 72$) and Groningen ($N = 56$) groups did not differ significantly on age and sex. The Groningen cohort had poorer cognition, but lower parkinsonian motor rating, earlier motor disease stage, and shorter disease duration. There was a trend for slightly lower LED

among Groningen subjects relative to Michigan subjects at the follow-up visit.

### Subhead 2: system-level PD cholinergic subgroups at baseline
Imaging acquisition and preprocessing methods are elaborated upon in supplementary materials section 3. A schematic representation of the algorithm applied to regional cholinergic PET uptake data to derive biology-based PD cholinergic subgroups is shown in Fig. 1, and more detailed descriptions are provided in the methods section.

PCA was performed on Z-scored regional cholinergic PET uptake data as described in Fig. 1 and methods section. A total of 5 principal components covering the whole brain were retained based on the pre-specified eigenvalue cutoff of 1. After varimax rotation, all loading scores at or above the specified absolute-value threshold of 0.5 were included. Spatial distribution of component loadings over both cortical and subcortical regions is presented in supplementary materials section 4, Fig. S1. The first principal component had highest loadings in posterior cortices, including primary and association visual cortices (designation: posterior cortices). The second principal component had strongest loadings in centro-cingulate cortices, including paracentral, precentral, superior frontal, and posterior cingulate cortices (designation: centro-cingulate). The third principal component exhibited the most extensive subcortical involvement, strongest in striatum, thalamus, and amygdala, with weaker involvement in the insula and rostral anterior cingulate cortex (designation: limbic/subcortical). The fourth principal component included cerebellar hemispheres, vermis, and nuclei (designation: cerebellum). The fifth principal component was primarily related to FEOBV binding in entorhinal cortex (designation: entorhinal cortex).

Among 245 PD subjects at baseline, 29% ($N = 71$) were classified as hyper-cholinergic, 40.8% ($N = 100$) were classified as mixed-cholinergic, and 30.2% ($N = 74$) were classified as hypo-cholinergic. Among the hyper-cholinergic subjects, upregulation of the cerebellar component was most frequent (94.4%), followed by upregulation of limbic/subcortical (33.8%) and centro-cingulate (15.5%) components. The frequency of upregulation in posterior cortical (1.4%) and entorhinal (1.4%) components was negligible. Among the hypo-cholinergic subjects, deficits of the posterior cortical (82.4%) component were most frequent, followed by less frequent deficits of entorhinal (35.1%), centro-cingulate (32.4%), cerebellar (13.5%), and limbic/subcortical (8.1%) components. Among mixed-cholinergic subjects, upregulation of the cerebellar (13%) component was mainly present with only negligible proportion of upregulation in the subcortical (2%) component and no instances of upregulation in any of the other components. Among mixed-cholinergic subjects, deficits were most commonly observed in the posterior cortical (12%) component, and less commonly observed in the entorhinal (5%), centro-cingulate (3%), and limbic/subcortical (2%) components.

Group comparisons on clinical characteristics between the PD cholinergic subgroups at baseline are presented in Table 2. PD patients assigned to the mixed-cholinergic and hypo-cholinergic subgroups tended to at a higher motor disease stage and exhibited greater severity of motor symptoms than those assigned to the hyper-cholinergic subgroup.

A post-hoc reference region validation analysis was performed, to examine whether further analyses may be susceptible to a subgroup-specific bias in quantification. No evidence for subgroup-specific quantification bias was observed, but potential influence of PET scanner and age on reference region signal were observed in the post-hoc analyses, which informed the inclusion of these factors as covariates in further analyses (see supplementary materials section 5, Table S2, Fig. S2). An additional post-hoc analysis was performed examining whether appreciable structural brain atrophy may be present in the combined PD sample or individual PD cholinergic subgroups relative to controls, to support the use of partial volume

**Table 1 | Mean ( ± SD) values of baseline sample demographic and clinical variables of the combined Michigan and Groningen group, individual center values, and statistical comparisons between the two groups**

| Variable | Combined (N = 245) | Michigan (N = 149) | Groningen (N = 96) | Statistic | P |
|---|---|---|---|---|---|
| Sex male/female (Male % total) | 178/67 (72.6%) | 113/36 (75.8%) | 65/31 (67.7%) | $\chi^2 = 1.555$ | 0.2124 |
| Age (years) | 66.8 ± 8 | 67.21 ± 7.57 | 66.24 ± 8.61 | t = 0.907 | 0.3658 |
| MoCA score | 25.7 ± 3.2 | 26.2 ± 3.26 | 25 ± 3.04 | t = 2.919 | 0.0039* |
| MDS-UPDRS$_{II}$ total score | 8.3 ± 5.7 | 7.32 ± 6.08 | 9.88 ± 4.69 | t = −3.686 | <0.001* |
| MDS-UPDRS$_{III}$ total score | 35 ± 13.6 | 37.58 ± 13.98 | 30.98 ± 11.99 | t = 3.919 | <0.001* |
| MDS-UPDRS PIGD subscore | 0.2 ± 0.1 | 0.18 ± 0.16 | 0.13 ± 0.1 | t = 2.974 | 0.0032* |
| Hoehn & Yahr | 2.2 ± 0.7 | 2.46 ± 0.6 | 1.8 ± 0.61 | t = 8.251 | <0.001* |
| Motor disease duration (years) | 4.4 ± 4.3 | 6.04 ± 4.73 | 1.73 ± 1.11 | t = 10.616 | <0.001* |
| LED (mg) | | 603.47 ± 413.1 | | | |

Gender distribution is presented as proportions. Unequal variance independent samples t tests were applied to test group differences in numerical variables while chi-square contingency tests were used for categorical variables. Statistically significant site comparisons at uncorrected two-tailed α = 0.05 are marked with an asterisk.
*Group differences between the Michigan and Groningen cohorts.
*MoCA* Montreal Cognitive Assessment, *MDS-UPDRS* Movement Disorder Society Revised Unified Parkinson's Disease Rating Scale, *PIGD* postural instability and gait difficulty, *LED* levodopa equivalent dose.

correction (PVC) in subsequent voxel-wise analyses. Evidence of structural brain atrophy was observed, which was specific to the hypo-cholinergic subgroup, thus supporting the use of PVC in subsequent analyses (see supplementary materials section 6).

**Subhead 3: system-level PD cholinergic subgroup longitudinal transition**

Table 3 presents a longitudinal transition matrix between cholinergic subgroups. Chi-square contingency test on this transition matrix demonstrated statistically significant evidence ($\chi^2 = 87.750$, $p < 0.001$) for a strong (*Cramer's V* = 0.585) association between baseline and follow-up subgroup assignments. Hyper-cholinergic subjects were most likely to remain hyper-cholinergic at follow-up (68.4%), less likely to transition into the mixed-cholinergic (29%), and unlikely to transition into hypo-cholinergic subgroup (2.6%). Mixed-cholinergic subjects were most likely to remain in the mixed-cholinergic subgroup (75.9%), less likely to transition to the hypo-cholinergic subgroup (18.5%) and least likely to transition back into hyper-cholinergic subgroup (5.6%). Hypo-cholinergic subjects were most likely to remain hypo-cholinergic (63.9%), less likely to transition into mixed-cholinergic (33.3%), and least likely to transition to hyper-cholinergic subgroup (2.8%). Taken together, these findings demonstrate that system-level subgroups assignments are longitudinally stable, with a clear pattern of transition from mostly upregulation of the cholinergic system, to a mix of upregulation and downregulation, and finally to mostly downregulation.

**Subhead 4: voxel-level PD subgroup comparisons against controls**

Voxel-wise thresholded (FDR-corrected at $P < 0.05$; controlling for age and scanner, see supplementary materials section 5) statistical parametric maps (SPM) corresponding to baseline cholinergic subgroup comparisons against the pooled control participant sample are presented in Fig. 2. The hyper-cholinergic subgroup ($N = 71$) showed higher [$^{18}$F]FEOBV binding relative to controls across cerebellar hemispheres and vermis, anteroventral striatum, putamina, mesencephalon, bilateral metathalamus (medial and lateral geniculate nuclei), thalamus (primarily dorsomedial, pulvinar, and ventral posterior lateral nuclei), fimbria, hippocampus, right more than left amygdala, right more than left gyrus rectus, anterior and inferior cingulum, and basal forebrain. [$^{18}$F]FEOBV binding deficits were statistically significant in posterior (occipito-parietal) cortices. The mixed cholinergic subgroup ($N = 100$) showed no statistically significant evidence of higher than normal [$^{18}$F]FEOBV binding. A more extensive topography of cholinergic deficits is observed in the mixed-cholinergic group than in the hyper-cholinergic group, with additional involvement of mid-to-superior temporal and prominent retro-splenial cortices. The hypo-cholinergic subgroup ($N = 74$) did not show statistically significant evidence of cholinergic upregulation relative to control participants and exhibited the most extensive topography of cholinergic deficits. The deficits in posterior occipito-parietal and more global temporal cortices were more severe in the hypo-cholinergic than in the other two subgroups, and additional deficits were observed in frontal cortices, thalami (right thalamus relatively spared), caudate nuclei, and limbic and paralimbic (insula, opercular) regions.

Post-hoc cross-sectional voxel-wise comparisons against the combined sample of controls were performed on site-specific subsets of data (Michigan-only, Groningen-only), with and without partial volume correction (PVC), to confirm the robustness of the observed topographies (see supplementary materials section 7). Despite clinical, motor, cognitive, and duration of disease, and dopaminergic treatment differences between the de novo Groningen cohort and the more advanced Michigan cohort, the overall topography of cholinergic upregulation and downregulation was reproduced across these subset analyses. The topography of cholinergic upregulation was successfully reproduced in the more advanced Michigan cohort among the hyper-cholinergic patients (Fig. S3). Relatively more pronounced cholinergic upregulation and spared cholinergic integrity across subgroups in the de novo Groningen cohort (Fig. S4). Re-analysis without PVC still reproduced the overall topography but adversely affected the spatial specificity of the obtained findings (Figs. S5–7). Lastly, we performed a confirmatory analysis of voxel-level group comparison findings with system-level principal component scores used to define the subgroups (see supplementary materials section 8). The system-level group comparison findings broadly agree with those observed on the voxel-level, suggesting the most widespread topography of upregulation in

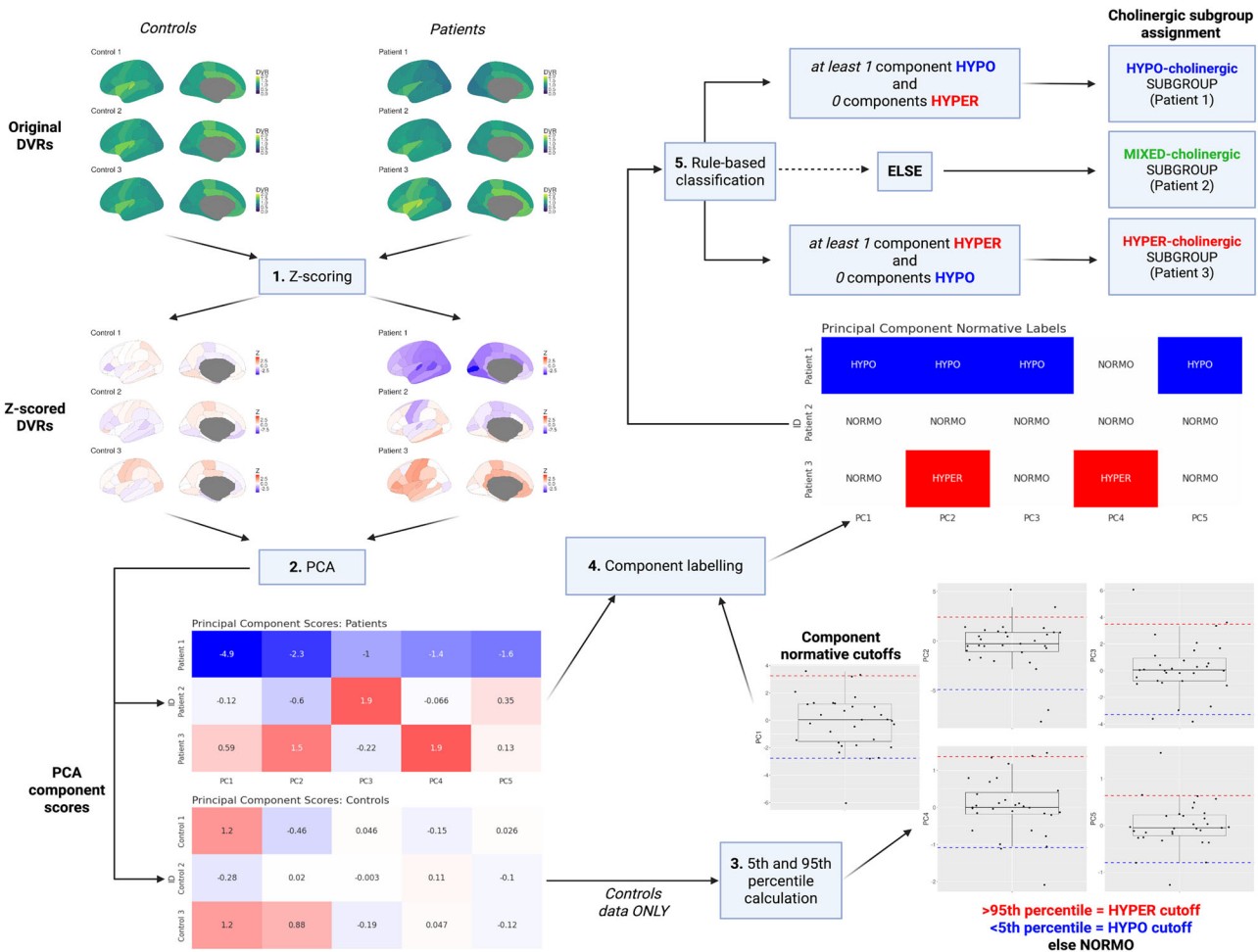

**Fig. 1 | Schematic representation of cholinergic subgroup biology-based classification algorithm.** Step 1 involves the Z-scoring of raw patient data (DVR, distribution volume ratio) relative to controls, with adjustment for effects of age and sex among controls (white: normative level, red: higher than normative, blue: lower than normative). Step 2 involves running a principal component analysis (PCA) on control and patient data, to obtain principal component scores. Step 3 involves using principal component scores as obtained among control participants to define normative component score cutoffs for hypo-cholinergic (5th percentile) and hyper-cholinergic (95th percentile) label assignment. Step 4 involves assigning normative labels to each patient's cholinergic system principal components based on normative cutoffs obtained from control participants in the previous step. Step 5 involves using the pattern of principal component labels obtained for each patient to assign them into a cholinergic subgroup with a rule-based classification heuristic. Created in BioRender. Roytman, S. (2025) https://BioRender.com/jk21glb. DVR distribution volume ratio, PCA principal component analysis.

the subcortical and cerebellar components of the hyper-cholinergic subgroup, loss of upregulation and more pronounced cholinergic downregulation in the posterior component of the mixed-cholinergic subgroup, and finally most widespread cholinergic downregulation in the hypo-cholinergic group (Table S3).

## Subhead 5: voxel-level within-subject longitudinal progression by baseline PD cholinergic subgroups

Voxel-wise thresholded (FDR-corrected at $P < 0.05$; controlling for age, scanner, and number of days between baseline and follow-up scans) statistical parametric maps (SPM) corresponding to within-subject paired $t$ test analysis of interval changes in cholinergic system by cholinergic subgroup are presented in Fig. 3. Across all three subgroups ($N_{HYPER} = 38$, $N_{MIXED} = 54$, $N_{HYPO} = 36$), only longitudinal loss of cholinergic system integrity was observed, no upregulation. The topography of cholinergic system changes closely matches the topography of cholinergic downregulation observed cross-sectionally relative to controls. The differential pattern of within-subject cholinergic losses by subgroup suggests that the greatest rate of cholinergic losses occurs earlier in the disease, accompanied by gradual deceleration as patients transition from hyper-cholinergic to mixed-cholinergic to hypo-cholinergic subgroups, with hypo-cholinergic patients showing minimal evidence of longitudinal decreases (most likely due to floor effect). This inference is further supported by post-hoc subset analyses performed by site, wherein the most extensive interval losses are observed among Groningen de novo patients with more recent symptom onset, as compared to more advanced Michigan patients (see supplementary materials section 9, Figs. S8, S9).

## Subhead 6: baseline global cholinergic system clinical associations

Three global (whole-brain) summary measures were defined based on extreme deviation (below 5th and above 95th percentile of controls) of individual image voxels in spatially normalized parametric PET relative to controls. Hyper-cholinergic voxel proportion ($V_+$) was defined to capture the spatial extent of cholinergic upregulation (number of voxels above the 95th percentile of controls divided by total number of voxels), hypo-cholinergic voxel proportion ($V_-$) was defined to capture the spatial extent of cholinergic downregulation (number of voxels below the 5th percentile of controls divided by total number of voxels), and cholinergic progression ($C$) was defined as a composite measure of the prior two, in order to capture a holistic, coarse-grained measure of

**Table 2 | Baseline demographic and clinical characteristic *ANOVA* group comparisons of mixed and hypo-cholinergic against the hyper-cholinergic subgroup**

| Variable | Hyper-cholinergic (N = 71) | Mixed-cholinergic (N = 100) | Hypo-cholinergic (N = 74) | Model P value |
|---|---|---|---|---|
| Age (years) | 65.66 [63.8, 67.53] | 67.55 [65.98, 69.12] | 66.99 [65.16, 68.81] | 0.3088 |
| MoCA (score) | 25.86 [25.1, 26.61] | 25.77 [25.13, 26.41] | 25.53 [24.78, 26.28] | 0.8155 |
| Disease duration (years) | 3.47 [2.41, 4.53] | 4.46 [3.61, 5.31] | 5.25* [4.26, 6.24] | 0.0552 |
| PIGD severity (score) | 0.14 [0.11, 0.17] | 0.17 [0.15, 0.2] | 0.18 [0.14, 0.21] | 0.2213 |
| MDS-UPDRS part II (total score) | 8.18 [6.85, 9.52] | 8.61 [7.48, 9.74] | 8.08 [6.75, 9.41] | 0.8107 |
| MDS-UPDRS part III (total score) | 30.84 [27.7, 33.98] | 35.37* [32.73, 38.01] | 38.53* [35.48, 41.59] | 0.0028* |
| Hoehn & Yahr (score) | 1.94 [1.79, 2.1] | 2.24* [2.11, 2.38] | 2.39* [2.23, 2.54] | <0.001* |

Estimated marginal means with 95% intervals for each variable are presented by subgroup (regression coefficients corresponding to group contrasts for which the confidence interval did not overlap with 0 are marked with an asterisk). Model P values are presented for the *ANOVA* group comparisons and were considered statistically significant at Bonferonni adjusted two-tailed α = 0.00714 (0.05 / 7 comparisons), in which case they were marked with an asterisk.
*MoCA* Montreal Cognitive Assessment, *PIGD* postural instability and gait difficulty, *MDS-UPDRS* Movement Disorder Society Revised Unified Parkinson's Disease Rating Scale.

**Table 3 | Longitudinal subgroups contingency table / transition matrix**

| Baseline Subgroup | Follow-up Subgroup | | | Total (Baseline) |
|---|---|---|---|---|
| | HYPER | MIXED | HYPO | |
| HYPER | 26 68.4% | 11 29% | 1 2.6% | 38 29.7% |
| MIXED | 3 5.6% | 41 75.9% | 10 18.5% | 54 42.1% |
| HYPO | 1 2.8% | 12 33.3% | 23 63.9% | 36 28.2% |
| Total (Follow-up) | 30 23.4% | 64 50% | 34 26.6% | 128 100% |

Rows represent subgroup assignment at baseline, columns represent subgroup assignment at follow-up. Percentages on the margins represent the proportions of participants assigned to each subgroup at either baseline (row margin) or follow-up (column margin). Percentages in the cells of the contingency table represent the percentage of participants assigned to a baseline subgroup that transition to a given follow-up subgroup. Median time between visits was 3 years (range: [2.9, 3.3] years) for the Groningen participants and 2 years (range: [0.7, 4.6] years) for the Michigan participants.
$\chi^2 = 87.750 \cdot df = 4 \cdot$ Cramer's $V = 0.585 \cdot p < 0.001$

bi-directional cholinergic system changes in PD (see methods subhead 6 for detailed definitions, and supplementary materials section 10, Fig. S10 for a schematic representation of the algorithm, Fig. S11 for visualization by subgroup over age). Supplementary analyses demonstrate that these global measures are well predicted by the principal components used to define the cholinergic subgroups ($R^2$: $V_+ = 0.686$, $V_- = 0.559$, $C = 0.540$; Table S4) and that they exhibit differential longitudinal interval changes by subgroup that agrees with greatest rate of cholinergic losses in hyper-cholinergic subgroup followed by mixed-cholinergic and minimal longitudinal changes in hypo-cholinergic patients (see supplementary materials section 10 for details).

Table 4 presents regression coefficients for cross-sectional post-hoc correlation analyses of global cholinergic system measures in relation to clinically relevant outcomes (see methods subhead 6 for details). Hypocholinergic voxel proportion ($V_-$) associates positively with PIGD symptom severity, turn duration and negatively with MoCA scores. Hypercholinergic voxel proportion ($V_+$) associates negatively with overall motor symptom severity. The composite cholinergic system progression ($C$) which captures the joint loss of hyper-cholinergic and gain of hypo-cholinergic voxels associates positively with overall motor symptom severity, PIGD symptom severity, turn duration, and negatively with overall cognition.

### Subhead 7: longitudinal cholinergic system progression clinical models

Table 5 presents regression coefficients for longitudinal *post-hoc* interval-change robust regression models predicting longitudinal rate of change in clinical outcome ($\Delta Y$) as a function of rate of change in the composite cholinergic progression summary statistic ($\Delta C$). Interval lengths between baseline and follow-up visits ($\Delta T$), baseline cholinergic system progression state ($C_{BL}$), and baseline clinical measure values ($Y_{BL}$) are included as covariates. More rapid cholinergic system progression between baseline and follow-up visits (greater $\Delta C$) is associated with more severe overall declines in cognition (greater decrease in MoCA score), more severe rate of deterioration in executive cognitive function (greater decrease in executive domain Z-score), greater decrease in gait speed, and greater increase in turn duration.

### Discussion

With the exception of etiologic genetic subtyping, subtyping approaches in PD are based on clinical criteria[3]. Relationships between proposed subtype clinical features and underlying pathophysiologic mechanisms, however, are largely unknown. This gap is an obstacle to identifying novel therapeutic targets[13]. Conventional assessments of neuropathology have not been useful in identifying relevant pathophysiologic correlates of clinical subtypes. In a post-mortem study of PD subjects defined as mild motor-predominant, intermediate, and diffuse malignant PD subtypes, Fereshtehnejad & Postuma (2017) found no group differences in Lewy body or Alzheimer pathology[13]. While these results may reflect end-stage floor effects, they point to a need to apply more neurobiologically based methods to identify salient pathophysiologic mechanisms.

Our prior in vivo cholinergic systems imaging research suggested that more severe cholinergic systems deficits associate with a "diffuse malignant" PD subtype characterized by marked PIGD motor features (falls, freezing of gait) and cognitive impairments[11]. Our prior work also suggested the presence of upregulation of cholinergic neurotransmission in early PD as a compensatory mechanism[22]. The presence of a severe motor and non-motor disability phenotype associated with cholinergic systems deficits and evidence of cholinergic systems compensations implies the existence of earlier stages with subgroups characterized by less severe cholinergic losses and/or

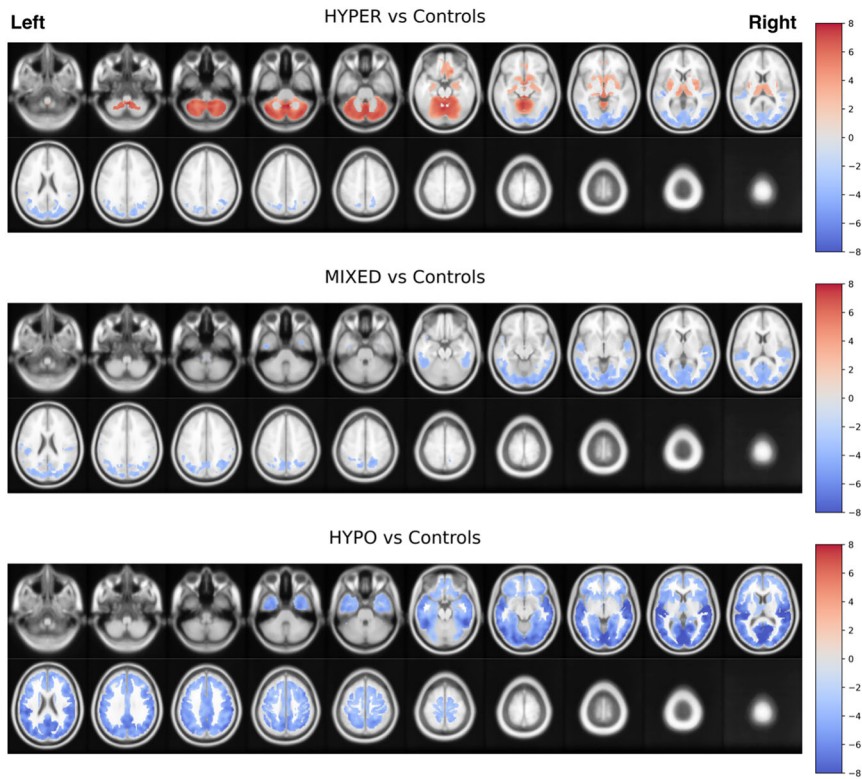

**Fig. 2 | Main voxel-based SPM analysis comparing each of the baseline cholinergic subgroups against control participants with independent samples *t* tests.** Statistical t-contrast images were thresholded with false discovery rate (FDR) threshold of $P < 0.05$. Positive contrast values (red color scale) indicate statistically significant upregulation of [18F]FEOBV uptake in subgroups relative to control subjects, with negative values indicating statistically significant deficits (blue color scale).

with initial relative masking of clinical deficits by cholinergic systems compensations. This pathophysiologic subgroup model predicts the existence of significant differences in cholinergic systems changes in earlier stage disease, that these differences are related to important clinical features, and that changes in brain cholinergic systems will parallel clinical progression. We explored this model in cross-sectional and longitudinal analyses of early to moderate stage PD.

### Subhead 1: cholinergic subgroups—frequencies and topographic signatures

Based on PCA analysis of regional [18F]FEOBV binding and a heuristic classification ruleset, we found evidence of subgroups defined by differential regional expression of cholinergic systems changes. Approximately 30% of subjects were categorized in the hyper-cholinergic subgroup, about 40% in the mixed-cholinergic subgroup, and about 30% in the hypo-cholinergic subgroup. Complementary cross-sectional whole brain voxel-based comparisons with normal controls revealed concordant patterns. Also corroborating the existence of cholinergic system subgroups are the distinctive regional topographies of VAChT upregulation and deficits. The hyper-cholinergic group generally had a distinctive topography of cerebellar hemispheres and vermis, anteroventral striatum, putamina, mesencephalon, bilateral metathalamus (medial and lateral geniculate nuclei), thalamus, fimbria, hippocampus, right more than left amygdala, right more than left gyrus rectus, anterior and inferior cingulum, and basal forebrain upregulation whereas posterior (parietal, occipital) cortices manifested cholinergic deficits in the hypo-cholinergic group. The voxel-wise cholinergic subgroup comparisons against normal controls confirmed these PCA-based findings but also revealed more intra-component regional topographic details. Within the hyper-cholinergic subgroup, for example, relatively greater upregulation

was found in the cerebellum, thalamus, metathalamus, and limbic archicortices. At the combined center analysis the mixed cholinergic subgroup demonstrated no evidence of significant upregulation but only significant hypocholinergic deficits in the posterior cortices and extending to the superior and posterior temporal cortices. More limited but significant upregulation in the mixed-cholinergic subgroup was only present in the less advanced Groningen PD subset in the cerebellum, lingual gyrus, fimbria, limbic archicortices, putamina, anteroventral striatum, right gyrus rectus and inferior aspect of the anterior cingulum. The combined 2-center hypocholinergic subgroup analysis demonstrate more diffuse cortical deficits in a posterior-to-anterior gradient, thalami, caudate nuclei, limbic and paralimbic cortices. It was only in the less advanced Groningen hypocholinergic group that mainly limited superior vermis and posterior mesencephic and diencephalic upregulation was observed.

These results imply a complex interplay of relatively selective degeneration within brain cholinergic systems in the context of likely compensatory changes across multiple cholinergic systems. Subcortical basal forebrain and pedunculopontine-laterodorsal tegmental (PPN-LDT) cholinergic projection neurons, medial vestibular (MVN) cholinergic projection neurons, and striatal cholinergic interneurons provide most of the brain cholinergic innervation. The predominant posterior cortical deficit topography of the hypocholinergic subgroup suggests preferential vulnerability of the cholinergic intermediodorsal (Ch4id) and posterior (Ch4p) subregions of the basal nucleus of Meynert (Ch4)[24]. Regions associated with the hypercholinergic topographic pattern - frontal lobe, cingulum, operculum, and amygdala receive projections from the anteromedial (Ch4am) and anterolateral (Ch4al) Ch4 subregions. Selden et al. (1988) identified two major output pathways from the Ch4 group with the lateral pathway (capsular division) innervating the amygdala and temporal cortices, and

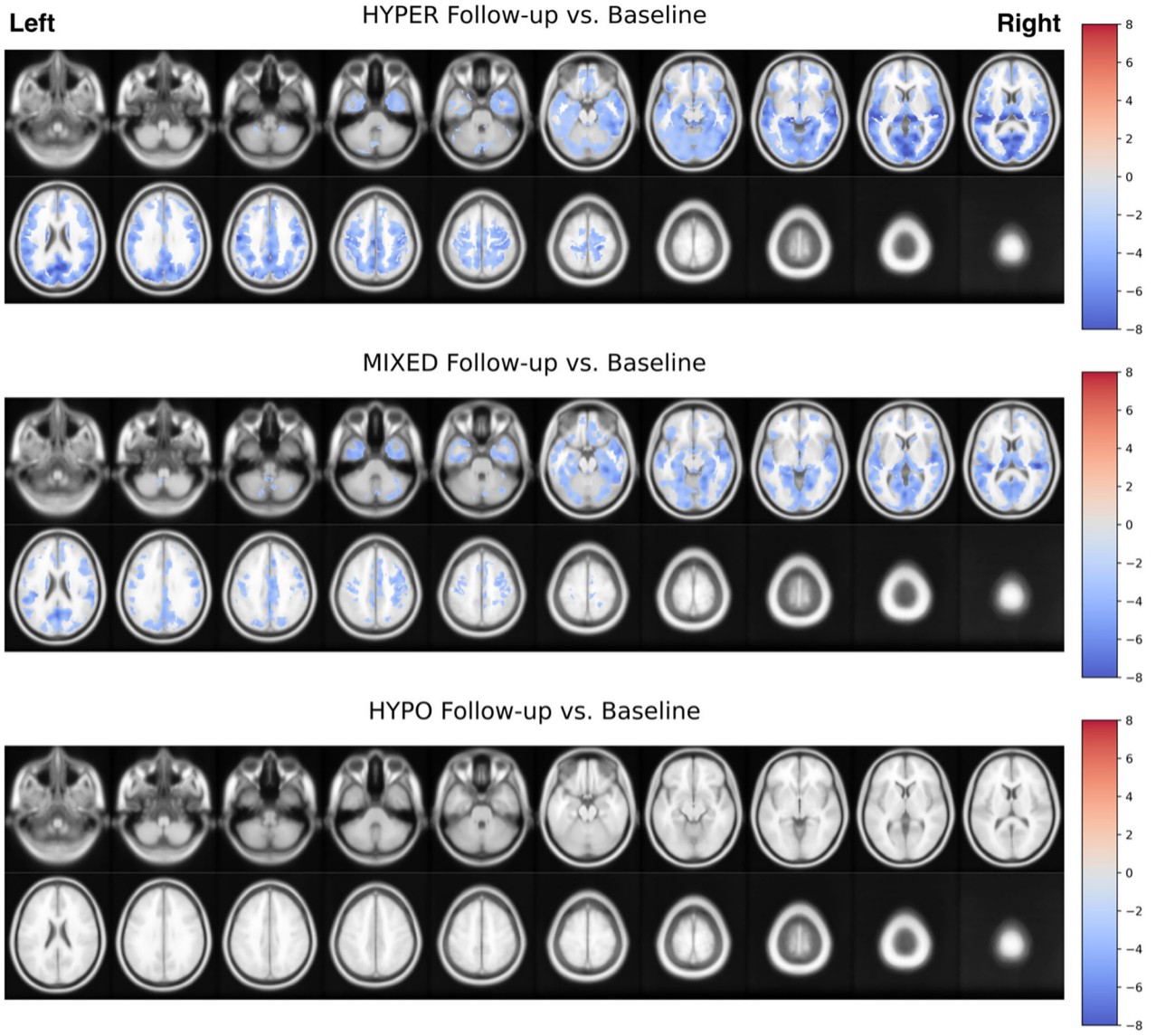

**Fig. 3 | Main voxel-based SPM analysis comparing each subgroup within-subject at follow-up visits relative to baseline visits with paired samples *t* tests (adjusted for days between visits, age, and scanner).** Statistical contrast images (positive and negative) were thresholded with false discovery rate (FDR) correction at $P < 0.05$. Negative (blue) contrast values indicate statistically significant down-regulation of [18F]FEOBV uptake in a given region relative to values observed at baseline.

the medial pathway innervating the gyrus rectus and anterior cingulum[25]. These differences may also contribute to pathway-specific vulnerability. Potential heterogeneity of vulnerability within the Ch4 group suggests that relative vulnerability of specific basal forebrain cholinergic neuron clusters may explain the posterior-to-anterior gradient of cholinergic denervation in PD rather than being the result of axonal length-dependent degeneration[26]. Cholinergic innervation is more dense in limbic than neocortical structures, with limbic and paralimbic cortices of the brain receiving denser cholinergic input from Ch4[27]. Anatomic connectivity and innervation density differences may reflect a higher biological reserve capacity in limbic cholinergic afferents.

In the hyper-cholinergic subgroup, regions innervated by the basal forebrain, PPN-LDT, MVN, and striatal cholinergic terminals exhibit increased [18F]FEOBV binding, consistent with widespread compensatory increased cholinergic neurotransmission. Consistent with this inference, a recent study found that Ch4 MRI density loss significantly correlated with resting-state functional connectivity

between the frontoparietal network and caudate nucleus, associated with compensatory recruitment of attention and executive cognitive functions during gait[28]. Increased cerebellar [18F]FEOBV binding was found in both the Michigan and Groningen hyper-cholinergic and the mixed and even hypocholinergic less advanced Groningen subgroups, suggesting altered cerebellar function as part of compensatory responses. Prior [18F]-fluorodeoxyglucose (FDG) brain PET studies, as well as cerebral perfusion studies across a spectrum of neurodegenerative disorders (Alzheimer disease, PD, dementia with Lewy bodies, REM sleep behavior disorder) show evidence of relatively upregulated cerebellar activity[29–33] and a recent [18F]FDG PET study using graph theory analysis found evidence of absolutely increased metabolic cerebellar activity[34]. These preliminary observations may suggest that compensatory increased cholinergic neurotransmission occurs in many regions and may play a role in modulating larger scale network changes in neurodegenerative disorders. Different constructs are proposed to explain compensatory processes, such as brain reserve or resilience[35]. These all remain theoretical constructs that need to be

**Table 4 | Cross-sectional post-hoc robust regression models examining the differential contributions of whole-brain level cholinergic deviation spatial extent measures to clinically relevant PD features**

| Outcome | N | Intercept | Independent contributions of upregulation and downregulation | | Contributions of combined deviation |
|---|---|---|---|---|---|
| | | | Hypo voxel prop. ($V_-$) | Hyper voxel prop. ($V_+$) | Cholinergic progression ($C$) |
| MDS-UPDRS Part II total | 243 | 7.845 [7.197, 8.492] | +0.669 [−0.03, 1.368] | +0.566 [−0.133, 1.265] | +0.284 [−0.369, 0.937] |
| MDS-UPDRS Part III total | 243 | 34.388 [32.676, 36.101] | +1.656 [−0.201, 3.513] | −2.286* [−4.143, −0.429] | +2.552* [0.757, 4.346] |
| MDS-UPDRS PIGD score | 245 | 0.142 [0.129, 0.154] | +0.024* [0.01, 0.037] | 0 [−0.014, 0.014] | +0.021* [0.008, 0.033] |
| MoCA score | 242 | 25.974 [25.6, 26.347] | −0.888* [−1.29, −0.486] | −0.296 [−0.698, 0.106] | −0.612* [−0.992, −0.233] |
| Executive function (normative Z) † | 138 | −0.531 [−0.71, −0.353] | −0.079 [−0.271, 0.112] | +0.063 [−0.129, 0.254] | −0.075 [−0.256, 0.106] |
| Memory function (normative Z) † | 138 | −0.359 [−0.507, −0.21] | +0.014 [−0.145, 0.173] | +0.077 [−0.082, 0.236] | −0.016 [−0.164, 0.133] |
| Velocity (m/s) † | 115 | 1.157 [1.119, 1.194] | −0.027 [−0.067, 0.014] | −0.015 [−0.055, 0.026] | −0.016 [−0.054, 0.022] |
| Cadence (steps/min) † | 115 | 106.625 [105.173, 108.078] | −1.302 [−2.863, 0.259] | −0.743 [−2.304, 0.818] | −0.992 [−2.486, 0.502] |
| Turn duration (s) † | 115 | 2.809 [2.698, 2.92] | +0.175* [0.055, 0.294] | +0.028 [−0.092, 0.147] | +0.156* [0.045, 0.266] |
| Double support time (% of gait cycle) † | 115 | 23.467 [22.704, 24.229] | +0.496 [−0.323, 1.316] | −0.249 [−1.068, 0.571] | +0.565 [−0.203, 1.333] |

Hyper ($V_+$) and hypo-cholinergic ($V_-$) voxel proportions represent the proportion of total voxels that exceed either of the respective voxel-level normative value thresholds and were included in a multivariate model predicting clinical outcome to examine the independent contribution of both hyper- and hypo-cholinergic brain changes. Cholinergic progression ($C$) is a composite measure that captures joint loss of hyper-cholinergic and gain of hypo-cholinergic voxels, and was used to predict clinical outcomes in a univariate regression model. Intercept represents the outcome measure mean value (with 95% confidence intervals) with respective regressors held at the sample mean. Strong regression coefficients (wherein the 95% confidence interval on the estimate does not overlap with 0) are marked with an asterisk. Outcome measures for which the analysis was performed only on Michigan patients due to data availability are marked with †.
*MDS-UPDRS* Movement Disorder Society Revised Unified Parkinson's Disease Rating Scale, *PIGD* postural instability and gait difficulty, *MoCA* Montreal Cognitive Assessment.

clearly defined. In this respect, cholinergic compensation may be mediated by interrelated changes in cholinergic systems and neural networks. For example, connectome-defined mechanisms of cholinergic compensation and dysfunction may also explain more regional cholinergic topographic heterogeneity based on the connectome disease propagation model where α-synuclein propagation is dependent on the ipsilateral connections that dominate connectivity of the human brain[36].

The concept of compensatory mechanisms in advanced disease is supported by older post-mortem studies reporting increased quantities and size of cholinergic nerve terminals in the substantia nigra pars compacta[37]. All these results are consistent with a 'compensatory' hypothesis that clinical effects of nigrostriatal losses may be initially masked by preserved to upregulated cholinergic activity of the cholinergic forebrain, striatum, or brainstem[38], with subsequent cholinergic neuron dysfunction/degeneration 'unmasking' the clinical motor and non-motor effects of nigrostriatal dopaminergic deficit[39]. Other studies support this hypothesis. Legault-Denis and colleagues reported increased [¹⁸F]FEOBV binding in the hippocampus in cognitively intact PD subjects but not in those with cognitive impairments[23]. A post-mortem study in mild cognitive impairment (MCI) due to Alzheimer disease found evidence of upregulation of choline acetyltransferase in the hippocampus and frontal cortex in early stage disease[40]. Superior frontal cortex choline acetyltransferase immunoreactive fiber and axon varicosity densities, however, were not altered in the MCI group but were significantly reduced in the group with dementia and correlated with impaired frontal lobe and global cognitive function[40].

**Subhead 2: longitudinal analyses of baseline defined subgroups**
Longitudinal analysis of baseline subgroups revealed evidence of complex evolution. The transition matrix analysis demonstrated significant associations between baseline and follow-up subgroup classifications, consistent with the presence of subgroups (Table 3). The relative proportion of subgroup classifications, however, shifted

somewhat. Longitudinal whole brain voxel-based analysis showed that the baseline-defined hyper-cholinergic subgroup demonstrated significant interval reductions involving predominantly the posterior parieto-occipital and temporal cortices along with less extensive subcortical-anterior cholinergic losses. In conjunction with the voxel-based and global spatial abnormality (hyper- and hypo-cholinergic voxel proportion) analyses, it is plausible that there is a progression from a hyper-cholinergic subgroup state through a mixed-subgroup state to a hypo-cholinergic subgroup state, the latter representing a form of pathologic end-stage disease. This would be consistent with prior work associating widespread cholinergic denervation with dementia. This interpretation is consistent with a model in which there are clinically salient subgroups in early to moderate disease with a final common denominator of widespread cholinergic (and other systems) systems degeneration.

**Subhead 3: differential clinical correlates of whole-brain global cholinergic system upregulation, downregulation, and net deviation measures**
Consistent with the pathophysiologic relevance of cholinergic subgroups is evidence of correlations between cholinergic terminal deficits and upregulation with important clinical features. Higher proportion of hypocholinergic voxels was associated with more severe motor (esp. PIGD motor features, slower turn duration) and more impaired cognition (Table 4). Higher proportion of hyper-cholinergic voxels appeared to differentially associate with less severe global parkinsonian motor rating independently of hypo-cholinergic voxel proportion. These associations with global hypo-cholinergic activity are consistent with our prior work[11] and add a higher level of specificity by quantifying the global spatial extent of extreme deviations from normative control values. This finding agrees with recent works demonstrating that cholinergic upregulation in the hippocampus appears to play a role in preserving normal cognition in PD subjects[22,23,41]. Longitudinal analyses (Table 5) further supports the

**Table 5 | Longitudinal post-hoc robust regression models examining the contributions of global spatial cholinergic system deviations progression to interval changes in clinically relevant PD features**

| Δ Outcome (ΔY) | N | Baseline Average | Δ Average (Intercept) | Baseline Outcome ($Y_{BL}$) | Baseline Progression ($C_{BL}$) | Years Since Baseline (ΔT) | Δ Progression (ΔC) | $R^2$ |
|---|---|---|---|---|---|---|---|---|
| MDS-UPDRS Part II total score | 128 | 7.781 [6.93, 8.632] | +0.982* [0.15, 1.814] | −1.749* [−2.633, −0.865] | +0.736 [−0.132, 1.604] | +0.308 [−0.588, 1.205] | +0.532 [−0.346, 1.41] | 0.111 |
| MDS-UPDRS Part III total score † | 67 | 32.082 [29.396, 34.768] | +7.77* [5.465, 10.075] | −1.888 [−4.293, 0.517] | +0.82 [−1.63, 3.27] | −0.16 [−2.521, 2.201] | +1.796 [−0.744, 4.335] | 0.022 |
| MDS-UPDRS PIGD sub-score † | 67 | 0.135 [0.11, 0.16] | +0.04* [0.019, 0.061] | −0.018 [−0.04, 0.004] | +0.013 [−0.01, 0.035] | −0.017 [−0.039, 0.005] | +0.021 [−0.002, 0.044] | 0.011 |
| MoCA score | 128 | 26.242 [25.732, 26.753] | +0.02 [−0.398, 0.439] | −0.831* [−1.264, −0.398] | −0.165 [−0.602, 0.271] | −0.153 [−0.615, 0.31] | −0.474* [−0.911, −0.036] | 0.119 |
| Executive function (normative Z) † | 67 | −0.577* [−0.852, −0.302] | +0.096 [−0.034, 0.226] | −0.128 [−0.262, 0.007] | −0.209* [−0.347, −0.071] | −0.026 [−0.159, 0.108] | −0.254* [−0.396, −0.112] | 0.079 |
| Memory function (normative Z) † | 67 | −0.274* [−0.485, −0.063] | +0.218* [0.081, 0.355] | −0.08 [−0.219, 0.06] | +0.042 [−0.104, 0.188] | −0.113 [−0.254, 0.028] | −0.145 [−0.293, 0.002] | 0.108 |
| Velocity (m/s) † | 64 | 1.268 [1.223, 1.312] | −0.01 [−0.035, 0.016] | −0.065* [−0.092, −0.038] | −0.004 [−0.032, 0.023] | +0.005 [−0.02, 0.029] | −0.027* [−0.054, −0.001] | 0.289 |
| Cadence (steps/min) † | 64 | 108.363 [106.41, 110.315] | +0.05 [−1.225, 1.324] | −3.077* [−4.394, −1.761] | +0.697 [−0.675, 2.069] | +0.844 [−0.354, 2.042] | −0.483 [−1.806, 0.839] | 0.319 |
| Turn duration (s) † | 64 | 2.746 [2.551, 2.94] | +0.493* [0.327, 0.66] | −0.394* [−0.618, −0.169] | +0.04 [−0.146, 0.225] | +0.075 [−0.075, 0.226] | +0.238* [0.071, 0.405] | 0.36 |
| Double support time (% of gait cycle) † | 64 | 23.347 [22.293, 24.401] | −1.787* [−2.559, −1.015] | −1.702* [−2.51, −0.894] | +0.4 [−0.423, 1.224] | −0.681 [−1.408, 0.047] | +0.267 [−0.519, 1.053] | 0.219 |

"*Baseline Average*" represents the unadjusted mean value for the outcome variable at baseline. "Δ *Average*" represents the mean change in the outcome measure from the baseline value, holding all other regressors at the sample mean. "*Baseline Outcome*" captures the effect of the outcome value at baseline on the change in the outcome value from baseline to follow-up. "*Baseline Progression*" captures the effect of baseline cholinergic system progression state (imbalance between hypercholinergic and hypocholinergic voxel counts scaled by their spatial extent) on longitudinal change in outcome measure. "*Years Since Baseline*" captures the effect of time interval between baseline and follow-up visits on change in the outcome measure independent of global cholinergic system progression. "Δ *Progression*" captures the effect of disease-related cholinergic system changes on interval change in the clinical outcome measure. Coefficient of determination ($R^2$) communicates the proportion of variance in outcome measure interval changes captured by the model. Outcome measures for which the analysis was performed only on Michigan patients due to data availability are marked with †. Strong regression coefficients (wherein the 95% interval did not overlap with 0) are marked with an asterisk.
*MDS-UPDRS* Movement Disorder Society Revised Unified Parkinson's Disease Rating Scale, *PIGD* postural instability and gait difficulty, *MoCA* Montreal Cognitive Assessment.

association between longitudinal rate of cholinergic system progression and important clinical features.

**Subhead 4: upregulation in longitudinal analyses**

In longitudinal combined 2-center analyses, the hypercholinergic subgroup only demonstrated significant cholinergic downregulation in a posterior-to-anterior cortical gradient, thalami, metathalamic, caudate nucleus, cholinergic forebrain, limbic and paralimbic regions. The 2-center combined mixed cholinergic subgroup analysis demonstrates only significant cholinergic downregulation but less severe compared to the longitudinal losses seen in the baseline-defined hypercholinergic subgroup. The 2-center analysis of the baseline defined hypocholinergic subgroup did neither show evidence of up nor downregulation likely suggested a denervation floor effect. Longitudinal analyses limited to the Michigan baseline-defined hypercholinergic and mixed cholinergic subgroups showed substantially less severe and less extensive cholinergic losses pointing to the presence of a cholinergic denervation floor effect with advancing PD. Interestingly, most cholinergic losses in the baseline-defined hyper and mixed cholinergic subgroups demonstrated substantially more cholinergic losses in a posterior-to-anterior gradient, consistent with a model that cholinergic losses are more prominent in early PD disease durations with a severe cholinergic denervation floor after about 5–6 years disease duration. The absence of significant longitudinal cholinergic upregulation in neither Michigan or Groningen cohort point to a temporal manifestation of cholinergic upregulation present during the first 2 years of motor symptomatic PD, at least at a group level. Future research is needed to investigate individual heterogeneity that may deviate from the group level-based estimates. Higher regional brain acetylcholinesterase activity was observed in Leucine Rich Repeat Kinase

2 (LRRK2) mutation carriers at risk of PD[42] pointing to cholinergic upregulation well before symptomatic motor manifestation.

**Subhead 5: limitations**

We only used a single presynaptic cholinergic biomarker (VAChT) and did not have information about post or presynaptic cholinergic nicotinic or muscarinic receptors that play a key role in cholinergic neurotransmission. While the de novo Groningen cohort exhibited features of earlier disease (lower MDS-UPDRS part III motor scores, lower H&Y stages), this group was approximately matched with the Michigan cohort in terms of age and exhibited slightly worse mean cognition and greater impact of motor symptoms on activities of daily living at baseline evaluations at this center but were otherwise very comparable. The MoCA and H&Y scores of Groningen patients with longitudinal observations hardly changed at the follow-up visit however. There were clinical and duration of disease differences between the Michigan and Groningen study site because of the de novo subject recruitment in Groningen where higher rates of hypercholinergic subgroup classification were observed compared to the longer duration of disease and predominant dopaminergic medication treated subjects in Michigan. The incorporation of data from both centers however was critical for offering a more comprehensive characterization of cholinergic system progression in PD including the earlier stages wherein upregulation relative to controls is more likely to be observed. Analyses were adjusted for the confounding effect of age, and we were able to reproduce the topographies of both upregulation and down-regulation in post-hoc analyses limited to only Groningen or Michigan patients. Longitudinal analyses were additionally adjusted for the differences in duration of longitudinal follow-up between the two centers. The two centers used different PET cameras, but all original imaging data from the Groningen data were analyzed in Michigan

using identical PET reconstruction and imaging processing steps. While the reference region approach for quantifying PET data may suffer from quantification bias related to uncontrolled for differences in data collection, we performed detailed analyses to ensure that quantification bias related to systematic variation in reference region region signal (by subgroup, site, visit, scanner, and age) had minimal effect on our findings (presented in supplementary materials section 5). Another potential limitation is that subjects with dementia (typically treated with cholinesterase inhibitors) or who were on neuroleptic drugs for psychosis were not eligible for the study. Given the coincidence of more severe PIGD motor features and dementia in the so-called 'malignant' cholinergic subgroup, this may result in a relative dilution of the clinical effects of cholinergic deficits. It should also be noted that posterior cortical hypocholinergic brain regions do not purely reflect loss of cholinergic nerve terminals but also may represent more widespread neurodegenerative or comorbidity changes rather solely the intrinsic loss of cholinergic nerve terminals[43]. Although there are important advantages from a pure biology-driven subgrouping approach, information about important disease progression modifiers, such as exercise, diet or medical comorbidities, should also be investigated in future studies and how this may differentially interact with specific cholinergic subgroups. Neurotransmission regulation is complex and probably involves other primary neurotransmitter defined systems, such as serotoninergic and or noradrenergic projections, that also contribute to clinical disease manifestation in PD[44]. Lastly, compensation likely involves multiple biological systems at multiple networks or regions not only in the brain but also in the peripheral nervous system. Despite these limitations, the correlations between clinical measures and global spatial extent of upregulated vs. reduced vesicular acetylcholine transporter topography quantified in this study indicated that bidirectional cholinergic system changes likely have clinical relevance in PD. Elucidation of mechanisms underlying cholinergic upregulation may inform a new therapeutic research strategy in PD.

## Methods

This study was approved by the Institutional Review Boards of the University of Michigan School of Medicine, Veterans Affairs Ann Arbor Healthcare System, and the University of Groningen Medical Center. Written informed consent was obtained from all subjects. Clinicaltrials.gov identifiers were NCT05459753 and NCT02458430. Informed consent was obtained from all subjects consistent with the Declaration of Helsinski. This 2-center study involved 245 non-demented PD subjects (Michigan $n = 149$, Groningen $n = 96$) and 31 control subjects (Michigan $n = 26$, Groningen $n = 5$). All subjects underwent [18F]FEOBV scanning at baseline, and a subset ($N = 128$, Michigan $n = 72$, Groningen $n = 56$) underwent [18F]FEOBV scanning at follow-up visit. Detailed description of imaging methods and preprocessing procedures is available in supplementary materials section 1. Median duration of time between baseline and follow-up was 3 years (range: [2.9, 3.3] years) for the Groningen subjects and 2 years (range: [0.7, 4.6] years) for Michigan subjects. Clinical and demographic characteristics were collected at both visits. Mean levodopa equivalent dose (LED) was computed based on patient-reported prescribed medications list[45]. Subjects underwent a motor and cognitive evaluation battery at both visits. The Movement Disorder Society-Revised Unified PD Rating Scale (MDS-UPDRS) motor examination was performed in the morning in the dopaminergic medication 'off' state for both cohorts at baseline, and in the 'on' state for the Groningen cohort only in the de novo subjects who received dopaminergic treatment at follow-up[46]. Subjects completed the Montreal Cognitive Assessment[47]. More detailed neuropsychological battery and instrumental gait assessment data were analyzed for the Michigan cohort. Data from patients from both sites has been previously published on[22,48–53].

## Subhead 1: subjects

PD subjects met the UK Parkinson's Disease Society Brain Bank clinical diagnostic criteria[54]. Typical striatal dopaminergic deficits were confirmed with PET imaging ([11C]DTBZ at Michigan; [18F]FDOPA at Groningen). The Groningen cohort consisted of de novo PD subjects at study entry with subsequent dopaminergic therapy[55]. The Michigan cohort consisted of mild-moderate PD subjects. Subjects with evidence of large vessel stroke or other intracranial lesions on anatomic imaging were excluded. At baseline, 41 Michigan PD subjects were taking a combination of dopamine agonist and carbidopa-levodopa medications, 82 were using carbidopa-levodopa alone, 9 were taking dopamine agonists alone, and 17 were not receiving dopaminergic therapy. No subjects were treated with anti-cholinergic or cholinesterase inhibitor drugs.

## Subhead 2: normative Z-scoring procedure

Primary analysis was performed on a standard FreeSurfer atlas regional [18F]FEOBV DVR values from 43 bilaterally averaged regions in addition to 3 cerebellar regions (vermis, hemispheres, nuclei) as defined from the SUIT cerebellar atlas. To covary out the effects of normal aging and sex on regional [18F]FEOBV DVR values, a linear model was fitted for each region on a dataset of 31 normal controls (NC; Michigan: 26, Groningen: 5), predicting the regional DVR from age and sex (see Eq. (1) below, $\varepsilon_{NC}$ represents model residual or error term):

$$Y_{NC} \sim \beta_{Age} + \beta_{Sex} + \varepsilon_{NC} \tag{1}$$

The sample of controls for fitting this normative model and subsequent analyses was curated by ensuring that their MoCA scores were at or above 25, and that no substantial deviations from normative values were observed on any of the cognitive domain composite measures. Normative regional [18F]FEOBV DVR Z-scores corrected for effects of age and sex among controls ($Z_{DVR}$) were obtained by subtracting from each observed regional value ($Y_{obs}$) the prediction of the NC model based on a participant's age and sex ($\hat{Y}_{NC}$) and dividing it by the standard deviation of control model residuals (std($\varepsilon_{NC}$)), see Eq. 2 below:

$$Z_{DVR} = \frac{Y_{obs} - \hat{Y}_{NC}}{std(\varepsilon_{NC})} \tag{2}$$

The resulting value represents the directional extent of deviation of the observed value from that which would be expected from control participants of the same age and sex. This procedure was applied to all control and PD participant regional [18F]FEOBV DVR values to ensure the effect of age and sex as observed among normal controls was covaried out. The described regional Z-scoring procedure was implemented in-house with *Python* open-source *statsmodels* module.

## Subhead 3: whole brain volume-of-interest PET analysis to define cholinergic subgroups

The two-center dataset of 245 PD subjects' baseline [18F]FEOBV scans (volume of interest extracted distribution volume ratio (DVR)) was used for principal components analysis (PCA). Regional DVR values were normatively Z-scored prior to PCA as described in previous section. The number of retained principal components was determined by an eigenvalue cutoff of 1. The resulting component loading matrix was varimax-rotated and thresholded based on an absolute-value cutoff of 0.5, with all loadings below that cutoff set to 0. The regional DVR data matrix was multiplied with the varimax-rotated and thresholded component loading matrix to yield the component scores matrix. Normative 5th and 95th percentile cutoffs were obtained for each of the principal components based on the component scores derived from

the 31 NC participants. Percentile cutoffs were used to assign cholinergic system component extreme deviation labels to each principal component for all PD participants. If a component score fell above the 95th percentile, the "hyper-cholinergic" label was assigned to that component, indicating that [18F]FEOBV binding is high relative to controls. If a component score fell below the 5th percentile cutoff, it was assigned a "hypo-cholinergic" label, meaning that [18F]FEOBV binding for a given principal component is low relative to NC.

Individual subjects within the baseline PD dataset were categorized based on cholinergic system principal component extreme deviation labels using an exclusive partition ruleset. The ruleset defines three categories of whole-brain cholinergic system deviation subgroups: hyper-cholinergic, mixed-cholinergic, and hypo-cholinergic. Subjects were categorized under the *hyper-cholinergic* subgroup if at least one of their principal components was assigned a hyper-cholinergic label and no other principal components were assigned a hypo-cholinergic label. Subjects were categorized under the *hypo-cholinergic* subgroup if at least one of their principal components was assigned a hypo-cholinergic label and no other principal components were assigned a hyper-cholinergic label. All other subjects were assigned to a *mixed-cholinergic* category, consisting of individuals with no principal components outside the 5th and 95th cutoffs or both hypo-cholinergic and hyper-cholinergic component labels present simultaneously. All PCA analyses, normative cutoff calculations, and ruleset-based classification procedures were implemented in-house as a set of *SAS* programming language macros.

### Subhead 4: longitudinal subgroup stability analysis

To identify whether a systematic pattern of transitions between subgroups could be discerned, a contingency table of transition frequencies was constructed, with baseline category for each participant represented as rows, and follow-up category represented as columns. Chi-square contingency test was used to test whether dependence existed between baseline and follow-up states, with inference of statistical significance of associations made at an α level set at 0.05. An open-source *R* programming language *sjPlot* library was used to implement the contingency table analysis.

### Subhead 5: baseline voxel-level comparisons between cholinergic subgroups and controls and voxel-level within-subject longitudinal change comparisons

Whole-brain voxelwise [18F]FEOBV PET baseline group comparisons were performed on a sample of 245 PD participants to compare each cholinergic subgroup to a sample of 31 normal controls with independent samples *t* tests using SPM12 software. Statistical parametric maps were thresholded at false discovery rate (FDR) corrected threshold of $P < 0.05$ with minimum cluster size of 50 voxels. Confoundng effects of age and scanner were included as covariates.

A follow-up multi-site PD dataset of 128 subjects (Michigan $n = 72$, Groningen $n = 56$) was used to study the nature of cholinergic systems progression. Longitudinal interval changes were assessed with whole brain paired samples *t* test within each of the 3 cholinergic subgroups defined at baseline, adjusted for duration between baseline and follow-up visits, age at baseline, and scanner. Statistical parametric maps were thresholded at false discovery rate (FDR) corrected threshold of $P < 0.05$ with minimum cluster size of 50 voxels.

### Subhead 6: post-hoc analysis to characterize the spatial extent of cholinergic upregulation and downregulation relative to controls using whole-brain summary statistics

Z-scoring relative to controls, normative cutoff calculation, and classification of individual gray matter voxels into hyper or hypocholinergic was performed on the spatially normalized parametric PET images in the same manner as described for the *VOI*-based analyses in the methods section above, except scanner was added as an additional

covariate into the normative Z-scoring model given our findings presented in supplementary materials section 5. A schematic illustration of the algorithm used to extract hyper- and hypo-cholinergic voxel counts from spatially normalized [18F]FEOBV parametric PET images is provided in Fig. S10 of the supplementary materials section 10. Proportion of hypercholinergic ($\frac{HYPER}{TOTAL}$) and hypocholinergic ($\frac{HYPO}{TOTAL}$) voxels was obtained by dividing their respective counts by the total number of gray matter voxels ($TOTAL = 122317$). Z-scoring of image voxels relative to controls was implemented in-house using open-source *Julia* programming language. Calculation of whole-brain cholinergic system measures was performed using open-source *Python* programming language and *ANTsPyX*, *pandas*, and *numpy* modules.

Hyper- and hypo-cholinergic voxel counts were also used to define a global cholinergic system progression measure. Global cholinergic system progression is defined as the difference between hypocholinergic and hyper-cholinergic voxel counts, divided by the difference between the total number of voxels and the number of extreme voxels (representing the number of remaining voxels not exceeding either the HYPO or HYPER threshold), see Eq. ( 3) below:

$$C = \frac{(HYPO - HYPER)}{TOTAL - (HYPO + HYPER)} \tag{3}$$

The resulting measure captures the magnitude of global imbalance between hypo-cholinergic and hyper-cholinergic voxel counts (numerator term) and scales it to the spatial extent of extreme deviation from normative values (denominator term). The measure is oriented in the direction of longitudinal cholinergic system progression, such that values less than 0 indicate early disease progression marked by predominantly hyper-cholinergic voxels, and values greater than 0 indicate late disease progression marked by predominantly hypo-cholinergic voxels. The normalizing term in the denominator represents the number of voxels remaining within the normative limit and is meant to emphasize the imbalance between hypo-cholinergic and hyper-cholinergic voxel counts when the spatial extent of cholinergic abnormality is widespread (when the number of remaining normal voxels is small), while de-emphasizing the imbalance when it's driven by relatively few abnormal voxels.

An exploratory analysis was performed to obtain better insight into potential clinical associations of global cholinergic system measures for the baseline cross-sectional dataset (including either participants from both sites or only from Michigan depending on data availability, N and sites involved in fitting each model are specified in Table 4). Examined measures included MDS-UPDRS part II, and III total scores, PIGD subscores (sum of MDS-UPDRS items 2.12, 2.13, 3.10, 3.11, and 3.12 divided by maximal possible score of 20), MoCA scores, cognitive domain z-scores (executive and memory), and gait characteristics as measures via APDM iSAW protocol (velocity, cadence, double support time, and turn duration). To examine the differential contribution of global cholinergic upregulation and downregulation to each of these clinically relevant measures, a set of multivariate robust regression models were fitted with hyper-cholinergic ($V_+$) and hypocholinergic ($V_-$) voxel proportions as independent regressors. To examine the contribution of joint loss of hyper-cholinergic and gain of hypo-cholinergic voxels as captured by the cholinergic progression ($C$) composite variable, the same set of models were fitted with $C$ as the only independent regressor. Regression model coefficients for the clinical association models are presented in Table 4.

To examine how the longitudinal rate of change in the global cholinergic progression measure ($\Delta C$) associated with the longitudinal rate of change in a set of clinically relevant outcome parameters, robust regressions were fitted on a sample of patients with both baseline and follow-up observations (including either participants from both sites or only from Michigan depending on data availability, N and sites involved in fitting each model are specified in Table 5). For a

given clinical outcome measure ($Y$), the interval change robust regression model was parameterized as such (see Eq. ( 4 ) below):

$$\Delta Y \sim Y_{BL} + C_{BL} + \Delta T + \Delta C \tag{4}$$

Where $\Delta Y$ represents interval change in an outcome measure from baseline to follow-up visit, $Y_{BL}$ represents baseline value observed for the outcome measure, $C_{BL}$ represents baseline cholinergic system progression state, $\Delta T$ represents the length of the time interval between baseline and follow-up visits, and $\Delta C$ represents change in the cholinergic system progression state from baseline to follow-up.

Unstandardized ($B$) regression coefficients with their corresponding 95% confidence intervals ($CI_{95}$) and $p$ values and adjusted model coefficients of determination ($R^2$) were reported. For cross-sectional and longitudinal combined Michigan and Groningen dataset post-hoc regression analyses, estimated marginal means for site, subgroup, and visit related categorical splits are given with 95% confidence intervals to enhance the interpretation of regression findings. For the cross-sectional clinical correlation analyses, intercept term coefficients ($B_0$) are given to inform about the sample mean value for the relevant clinical outcome with hypercholinergic and hypocholinergic voxel proportion held at the sample mean value. Strong regression coefficients for which the 95% confidence interval did not overlap with 0 were emphasized in both Tables 4 and 5 with an asterisk. All post-hoc regression analyses and plotting procedures were implemented in the *R* programming language using *MASS* (rlm) open-source package for regression modeling, *emmeans* for computing estimated marginal means from the fitted models, and *sjPlot* for regression model table generation.

### Reporting summary
Further information on research design is available in the Nature Portfolio Reporting Summary linked to this article.

## Data availability
The data that support the findings of this study are available from the corresponding author upon request.

## Code availability
All code which was relevant to the implementation of custom methods utilized in the present analysis is made available as a CodeOcean capsule: https://codeocean.com/capsule/3508354/tree.

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

## Acknowledgements

We thank study participants, cyclotron operators and study staff for their help with this study. Funding for the present work was received from the following grants: National Institutes of Health grant: P50 NS091856 (RLA, NIB) National Institutes of Health grant: P50 NS123067 (RLA, NIB) Parkinson's foundation (RLA, NIB) National Institutes of Health grant: R01 AG073100 (NIB) Department of Veterans Affairs: I01 RX001631 (NIB) Department of Veterans Affairs: I01 RX003397 (NIB) Galen + Hilary Weston foundation (TvL).

## Author contributions

Conceptualization: N.I.B. Methodology: N.I.B., S.R., K.A.F., P.J.H.S., T.v.L., R.A.K., P.K. Investigation: N.I.B., S.R., P.K. Visualization: S.R. Funding acquisition: R.A.K., N.I.B., T.v.L. Project administration: R.A.K., N.I.B., T.v.L. Supervision: R.A.K., N.I.B., T.v.L. Writing—original draft: N.I.B., S.R. Writing—review & editing: N.I.B., S.R., S.Z., G.C., F.M., A.L., S.S., R.L.A., P.K.

## Competing interests

The authors declare no competing interests.
