## [Transparent Peer Review file · Nature Communications]

A Multicenter Longitudinal Study of Cholinergic Subgroups in Parkinson Disease

Corresponding Author: Dr Nicolaas Bohnen

Version 0:

Reviewer comments:

Reviewer #1

(Remarks to the Author)

Thank you very much for submitting this manuscript about a very interesting original research study, which I have enjoyed reading. The work hypothesis is novel and pertains to a very hot topic of research, and the results may help advancing in the knowledge of the biological differentiation of Parkinson's disease (PD).

The methods used are sound and clear, and well-described in both the main manuscript and its supplementary materials so to allow for repetition of the experimental design. The mitigation process of the use of different PET and MRI scanners has also been well described. The use of two independent cohorts undergoing the same activities reinforces the strength of the results. The seemingly high MoCA and Hoehn and Yahr of the de novo PD group (Groningen cohort) has also been well discussed in the discussion section – it could be also discussed (as a further reassuring factor) that, after an average 3-year follow-up the mean values of these two parameters in the Groningen group barely changed.

I have the following few questions to ask to the Authors:

1. In the supplementary methods, the Authors report implementing Partial Volume Effects correction. I would like to ask whether the Authors decided to implement PVE correction in the light of (unreported) atrophy in the PD group compared to controls, given that age (another potentially confounding factor) is not difference across datasets. I would like the Authors to discuss the use of PVE correction in their datasets. In case of volumetric differences between the groups, these could be illustrated in a supplementary table.
2. I would like to ask about the choice of describing PD-related changes in cholinergic innervation starting from Principal Components (PCs) identified in the healthy control group. From previous studies PET studies exploring PC analysis of brain metabolism, or – with regards to neurotransmitter changes – Serotonergic innervation, disease-specific patterns have been described which are different from healthy controls and are related to disease severity, or progression. This has been touched in the subhead 1 of the discussion, and indeed, the finding in the present work that single groups of PD patients belonging to the same subgroup (e.g. hypercholinergic) display variable degrees of cholinergic alterations, may underlie the presence of disease-specific patterns, which would be different from those of healthy controls. What would be the advantage of using PCs starting from the healthy control group in terms of how would this explain PD-related changes?
3. Still on the topic, in the results section, subhead 3, I would like to ask whether the Authors have noted any PC-specific longitudinal change in cholinergic innervation in the PD groups, and whether changes in each of the PC (or any particular one) drive results more than others. This would be informative to understand which network could be more interested in PD patients for progression of disease.

(Remarks on code availability)

Reviewer #2

(Remarks to the Author)

The present study is a 2-center longitudinal PET study of vesicular acetylcholine transporter imaging in patients with Parkinson's disease (PD). A total of 248 non-demented patients with early to moderate PD and 31 healthy subjects were included. The study using a principal component analysis to identify 5 subgroups with cortical areas of increased or

decreased tracer binding. The longitudinal data with repeated scans within 2-3 years supported the hypothesis that increased binding is an early, subcortical phenomenon while decreased binding is a later, cortical feature.

Achieving a biological definition of PD is a major scope for the research community and will assist PD research in the future years. The present study aims to assess the non-dopaminergic neurotransmitter systems in PD, which is also highly relevant and could increase the understanding of the non-motor symptoms as well as potentially contribute to sub-classifications of the illness. Thus, the present study of cholinergic characterization of PD is relevant and the large cohort chosen seems to be a robust material. The use of [18F]FEOBV is a good choice for imaging cholinergic activity and is believed to reflect a combination of degeneration and possible up- and downregulation of the transporter. The use of scans from two sites could potentially affect the results but the authors have reliably matched the data analysis and used the site as a covariate to account for possible differences.

Overall, the study is of high standard. Some methodological concerns are, however, necessary to address:

The choice of a reference region approach increases the signal to noise substantially by removing some of the intersubject variance. On the other hand, the results may be sensitive to bias in case of changes in the reference region. The chosen reference region was supratentorial white matter, which is a robust choice as a cortical reference region and possibly cerebellar hemispheres could have biased the data. However, white matter changes are very common especially in patients with cognitive disturbances. Please report the values of the white matter region and possible change with age and subgroup. Please discuss the chosen reference region in your limitations and how changes here could impact the results.

In the voxelwise analysis, the Müller-Gärtner method was used for partial volume (PV) correction according to supplementary data. The Müller-Gärtner method is a well-established method using the segmentation of the MRI image into white matter, gray matter and CSF thereby allowing for separating reductions due to atrophy and reduction of the signal in the tissue. The corrections are often quite substantial especially for cortical gray matter due to the cortical thickness of only a few millimeters, which is much smaller than the scanner systems point-spread function. Thus, the method heavily relies on correct segmentation as well as coregistration and can be subject to bias and over-correction.

On the other hand, the authors chose to smooth the images after partial volume correction when entering the SPM voxelwise analysis, thereby efficiently removing the effect of the PV correction. Thus, the present results could despite partial volume correction possibly be attributed to atrophy – especially in the posterior cortical regions. Thus, the combined effect of the manipulations are obscure. Please also report data without PV correction to assess if the found results from the voxelwise analysis could result from overcorrection due to MRI segmentation flaws or if any effect was measurable. According to the method section (Subhead 2: Normative Z-scoring procedure) a regional analysis was applied without smoothing allowing for more robustly checking the results of the voxelwise analysis. However, it is unclear what the results of the regional analysis compared to the voxel-wise analysis was. Please clarify.

Table 3 with longitudinal data is quite convincing for the reliability of the results. Please include in the header the time interval (eg. Median + range) between baseline and follow-up.

Please state clearly if any of the included subjects has been included in previous publications.

The PCA found 5 significant groups. It is, however, unclear whether the deviation can be only positive or negative compared to healthy subjects or if both positive and negative deviations will be included in the same group? How does the method handle the much higher signal in centro-cingulate regions? The higher values here could skew the PCA to include these high-binding regions? What impact on the results do you think this variations in binding have?

As FDG and FEOBV has been found to correlate in AD patients (DOI: 10.1038/mp.2017.183), do you think you could have found the same results with FDG? If the found subtypes can indeed be replicated and used for future studies, the use of FDG would significantly increase the usefulness and availability of the biomarker.

(Remarks on code availability)

Reviewer #3

(Remarks to the Author)

This paper presents extensive analysis of cholinergic imaging data obtained from two large groups of PD subjects assessed at two different imaging centers. The main goal is to determine possible relations between cholinergic alterations and clinical features that could aid subtyping of PD. The results are obtained from a mixture of longitudinal and cross-sectional analysis using three different and possibly complementary approaches: (a) normative deviation system-level cholinergic covariance networks to define the subgroups, (b) voxel-level statistical parametric mapping comparisons to identify more granular regional expression patterns, and (c) whole-brain level summary statistics of cholinergic deviation spatial extent to capture the global trends of cholinergic system progression in PD.

While the approach is comprehensive, there are concerns about potential confounds affecting the results and the overall novelty of the findings compared to what is already published. Overall the paper is complex and it is hard to extract clear main messages. There is a lot of detailed information in the supplementary material, which is very hard to sift through without there being a clear summary of the relevance of each section; some of that material could be better presented in a chart form.

It is also hard to find clear new evidence of how cholinergic imaging contributes to subtyping over what is already known: for example, they find that the hypo-cholinergic subjects tend to be older, have poorer cognition and worse overall motor symptom severity, most of which is already known. An interesting observation is the finding that, as disease progresses, different regions show a different up/down-regulation trajectory, but there does not seem to be a relationship with clinical progression. Perhaps this aspect should be further expanded upon.

The merging of the two cohorts is of potential concern, even if many of them are addressed in the limitation section. The Groningen subjects are medication free at baseline, while the vast majority of the Michigan subjects are not. They have a higher UPDRS and lower Moca score in spite of a shorter disease duration (table 1). In addition, they are tested and scanned on medication at the follow-up visit, while the Michigan subjects are tested and imaged in the 'off' state. It is then hard to compare changes in their clinical metrics to those observed in the Michigan group. Could difference in the assessment/imaging protocol influence the findings presented in table 5? Further, the distribution of the data in Figs S4- S6 is quite different between the two groups (do they become more similar if the two groups are matched for disease duration?) and there did not seem to be a clear explanation for the difference.

Besides possibly extending the range of disease durations, what is thus the advantage of combining the two cohorts given these potential confounds? Several comparisons between the two cohorts are given in the supplementary material, but a clear explanation for their motivation/relevance does not seem to be given. The authors present a subset of the results from analysis that was only done using the Michigan data and claim that they are overall consistent with the analysis of the entire data set. While reassuring, this still does not guarantee that combining the data is completely valid.

Another concern is the very large number of comparisons – has any correction for multiple comparisons been made?

In summary, it is believed that the manuscript would greatly benefit from streamlining, making it more readable and highlighting the main findings and their novelty. The rationale for combining the two data sets should be clearly stated.

Specific comments

1. Introduction

When discussing the interpretation of the tracer binding, can the author comment on different results obtained with 11C-donepezil – AchE marker (see, for example Staer et al; EJ Neurol 2024)?

2. Methods

It is stated 'The Movement Disorder Society-Revised Unified PD Rating Scale (MDS-UPDRS) motor examination was performed in the morning in the dopaminergic medication 'off' state for both cohorts at baseline, and in the 'on' state for the Groningen cohort only in the de novo subjects who received dopaminergic treatment at follow-up'. This could introduce a bias when looking at the longitudinal changes in the clinical metrics.

3. Was there any difference in the baseline data between the subjects taking different medications (Michigan)?

4. Results

i. The Groningen group is described as having 'poorer cognition, greater impact of motor symptoms on activities of daily living, lower overall severity of motor symptoms, lower postural instability and gait difficulty (PIGD) symptom severity, lower tremor symptom severity, earlier motor disease stage, and lower disease duration from symptom onset than the Michigan cohort'

While these differences are addressed in the limitation section, one still wonders if they have a significant impact on the findings.

ii. 'There was a trend for slightly lower LED among Groningen subjects relative to Michigan subjects'. This needs to be clarified: presumably it refers to the follow-up visit as these subjects are described as de-novo at baseline.

iii. What was the VAF for each PC?

iv. The hyper/hypo cholinergic areas seem to vary within each subtype – can the authors comment on this? Is there a clinical correlate?

v. Fig 2 – on a group level the hypocholinergic group shows some hypercholinergic areas – this seems mildly at odds with defining these subjects hypocholinergic as per authors' definition (even though this is obtained from voxel-level analysis)

vi. Fig 3 – could the comparison of the follow-up scan be made wrt to controls again as confirmation of the changes observed between baseline and follow-up? Given the relatively short time interval between baseline and follow-up, there should still be relatively good age matching between PD and HC.

vii. Table 4 needs to be explained better. Are the results corrected for multiple comparisons?

viii. Supplementary figures (for example S5) the hyper group has several hypo voxels (up to 20%). Can the authors comment on this? Would plotting those data as a function of disease duration provide useful information?

ix. Table 5 – see comment on Delta-UPDRS above. Are any of the results adjusted for multiple comparisons? Which

regressions are overall significant? Some R2 values are very low.

(Remarks on code availability)

Version 1:

Reviewer comments:

Reviewer #1

(Remarks to the Author)

I thank the authors for considering my comments, especially with regards to further explanations on the use of PVC and providing more justifications for the use of it. I appreciate the new explanations and the figures in the supplementary material illustrating the results without using PVC.

I am personally okay with the explanations provided by the Authors.

(Remarks on code availability)

N/A

Reviewer #2

(Remarks to the Author)

The authors have sufficiently answered my concerns and I have no further recommendations.

(Remarks on code availability)

Reviewer #3

(Remarks to the Author)

The authors have appropriately addressed all my comments.

(Remarks on code availability)

made.

Reviewer Comments and Rebuttals

Reviewer #1 (Remarks to the Author):

Thank you very much for submitting this manuscript about a very interesting original research study, which I have enjoyed reading. The work hypothesis is novel and pertains to a very hot topic of research, and the results may help advancing in the knowledge of the biological differentiation of Parkinson's disease (PD). The methods used are sound and clear, and well-described in both the main manuscript and its supplementary materials so to allow for repetition of the experimental design. The mitigation process of the use of different PET and MRI scanners has also been well described. The use of two independent cohorts undergoing the same activities reinforces the strength of the results.

We thank the reviewer for many kind feedbacks on the manuscript!

0. The seemingly high MoCA and Hoehn and Yahr of the de novo PD group (Groningen cohort) has also been well discussed in the discussion section – it could be also discussed (as a further reassuring factor) that, after an average 3-year follow-up the mean values of these two parameters in the Groningen group barely changed.

We thank the referee for the comment. As suggested, a brief mention of this fact was added in the discussion section (subhead 5, limitations).

1. In the supplementary methods, the Authors report implementing Partial Volume Effects correction. I would like to ask whether the Authors decided to implement PVE correction in the light of (unreported) atrophy in the PD group compared to controls, given that age (another potentially confounding factor) is not difference across datasets. I would like the Authors to discuss the use of PVE correction in their datasets. In case of volumetric differences between the groups, these could be illustrated in a supplementary table.

Correction for partial volume effect was implemented in light of known issues with underestimation of tracer binding within and overestimation of tracer binding around high uptake areas (i.e. the striatum) for [¹⁸F]FEOBV. We performed a post-hoc analysis, evaluating whether there are appreciable differences in gray matter atrophy between both all PD participants in our sample and also between individual cholinergic subgroups with controls at baseline (see supplementary materials section 6). In the comparison by PD cholinergic subgroup we did observe that hypocholinergic PD patients exhibited a statistically significant higher positive discrepancy between brain age and biological age (greater estimated brain age than biological age), with a strong standardized effect size ($\beta=0.7$ [0.3, 1.1], $p=0.001$). Thus, though our initial

considerations for the use of PVC were not primarily informed by evidence of gray matter atrophy within our sample, our post-hoc analyses support the use of PVC, especially to mitigate partial volume effect in the subgroup most affected by cholinergic losses (hypocholinergic), due to more severe brain atrophy observed among them when appropriate measures are used. Additionally, we performed the voxelwise analyses with and without PVC, to evaluate the influence of this preprocessing step on our findings (see supplementary materials section 7).

2. I would like to ask about the choice of describing PD-related changes in cholinergic innervation starting from Principal Components (PCs) identified in the healthy control group. From previous studies PET studies exploring PC analysis of brain metabolism, or – with regards to neurotransmitter changes – Serotonergic innervation, disease-specific patterns have been described which are different from healthy controls and are related to disease severity, or progression. This has been touched in the subhead 1 of the discussion, and indeed, the finding in the present work that single groups of PD patients belonging to the same subgroup (e.g. hypercholinergic) display variable degrees of cholinergic alterations, may underlie the presence of disease-specific patterns, which would be different from those of healthy controls. What would be the advantage of using PCs starting from the healthy control group in terms of how would this explain PD-related changes?

The principal component analysis (PCA) used to define the subgroups was performed on a combined sample of 31 control patients and 245 PD patients at baseline, which means the resulting principal components would largely be driven by the greater number of PD participants, and thus capture the disease-related covariance patterns.

3. Still on the topic, in the results section, subhead 3, I would like to ask whether the Authors have noted any PC-specific longitudinal change in cholinergic innervation in the PD groups, and whether changes in each of the PC (or any particular one) drive results more than others. This would be informative to understand which network could be more interested in PD patients for progression of disease.

We performed a post-hoc analysis examining the longitudinal changes in the principal component scores on our sample of 128 PD patients with both baseline and follow-up scans (adjusted for age, scanner, and days between visits). All principal components demonstrated evidence of longitudinal cholinergic down-regulation (in agreement with our voxelwise within-subject longitudinal comparison, see figure 3), with the exception of the cerebellar component (PC4; $\beta=-0.01$ [-0.09, 0.08], $p=0.873$), which is in agreement with our finding of relatively more robust residual upregulation in the cerebellum observed in mixed-cholinergic and hypo-cholinergic subgroup of *de novo* Groningen PD

patients on the voxelwise comparisons against controls (see supplementary materials section 7). The strongest longitudinal down-regulation was observed in posterior cortices (PC1; $\beta=-0.24$ [-0.35, -0.12], $p<0.001$) and entorhinal cortex (PC5; $\beta=-0.28$ [-0.43, -0.12], $p=0.001$). Strong longitudinal downregulation in cholinergic activity in posterior cortices well-agrees with our prior work showing longitudinal posterior-to-anterior gradient of decrease in acetylcholinesterase hydrolysis of PD patients as measured via [11C]PMP PET (Bohnen et al. 2022). Our group has also previously shown that cholinergic losses in the entorhinal cortices play a key role in the development of more severe PIGD symptoms (Bohnen et al. 2021) suggesting that strong longitudinal downregulation in PC5 may be of clinical relevance.

Reviewer #2 (Remarks to the Author):

The present study is a 2-center longitudinal PET study of vesicular acetylcholine transporter imaging in patients with Parkinson's disease (PD). A total of 248 non-demented patients with early to moderate PD and 31 healthy subjects were included. The study using a principal component analysis to identify 5 subgroups with cortical areas of increased or decreased tracer binding. The longitudinal data with repeated scans within 2-3 years supported the hypothesis that increased binding is an early, subcortical phenomenon while decreased binding is a later, cortical feature.

We thank the reviewer for their thorough evaluation of our manuscript. The description of our analysis strategy is accurate. As we were going through revisions and taking a closer look at our data, 2 participants from University of Michigan and 1 participant from University of Groningen have since been excluded from the sample size, the former two on account of no evidence of dopaminergic denervation and taking acetylcholinesterase inhibitor, the latter one on account of later differential diagnosis of PSP. Otherwise, everything else summarized about our study is still accurate.

Achieving a biological definition of PD is a major scope for the research community and will assist PD research in the future years. The present study aims to assess the non-dopaminergic neurotransmitter systems in PD, which is also highly relevant and could increase the understanding of the non-motor symptoms as well as potentially contribute to sub-classifications of the illness. Thus, the present study of cholinergic characterization of PD is relevant and the large cohort chosen seems to be a robust material. The use of [18F]FEOBV is a good choice for imaging cholinergic activity and is believed to reflect a combination of degeneration and possible up- and downregulation of the transporter.

We believe that the reviewer summarized the overall scope and unique contributions of our work very well.

The use of scans from two sites could potentially affect the results but the authors have reliably matched the data analysis and used the site as a covariate to account for possible differences.

While site is not any longer used as a covariate in our analyses, we performed an additional supplementary analysis that we believe offers stronger evidence of the validity in combining the data from both sites, by showing that no quantification bias related to the reference region approach is present (see supplementary materials section 5). We believe that this approach is superior to including site as a covariate, since site inevitably encompasses a number of critical differences between the samples that may be closely related to relevant variation in the cholinergic system (*de novo* vs. advanced, disease duration, etc). Reference region uptakes however are assumed to not differ between sites for the reference region quantification approach to be considered valid, which is an assumption that we confirmed in the post-hoc supplementary analyses mentioned above.

Overall, the study is of high standard. Some methodological concerns are, however, necessary to address:

We thank the reviewer for their generous assessment and for the many constructive feedbacks that drastically enhanced the rigor and quality of our manuscript.

1. The choice of a reference region approach increases the signal to noise substantially by re-moving some of the intersubject variance. On the other hand, the results may be sensitive to bias in case of changes in the reference region. The chosen reference region was supratentorial white matter, which is a robust choice as a cortical reference region and possibly cerebellar hemispheres could have biased the data. However, white matter changes are very common especially in patients with cognitive disturbances. Please report the values of the white matter region and possible change with age and subgroup. Please discuss the chosen reference region in your limitations and how changes here could impact the results.

We thank the reviewer from pointing out this potential limitation, and gave it more explicit mention in the limitation section of our discussion. We added a post-hoc reference region standardized uptake value (SUV) comparison between subgroups relative to controls and within-subject longitudinally and were able to demonstrate that there were no statistically significant differences either between PD subgroups and controls or longitudinally (see supplementary materials section 5). Thus, while the reference region approach indeed suffers from limitations, the evidence does not appear to support the conclusion that our findings are driven by confounding variation in the reference region. However, the confounding influence of age and scanner heterogeneity that we identified in our post-hoc analysis were used to inform the inclusion of these factors as covariates in subsequent analyses, to further ensure that our results are not driven by these extraneous factors.

2. In the voxelwise analysis, the Müller-Gärtner method was used for partial volume (PV) correction according to supplementary data. The Müller-Gärtner method is a well-established method using the segmentation of the MRI image into white matter, gray matter and CSF thereby allowing for separating reductions due to atrophy and reduction of the signal in the tissue. The corrections are often quite substantial especially for cortical gray matter due to the cortical thickness of only a few millimeters, which is much smaller than the scanner systems point-spread function. Thus, the method heavily relies on correct segmentation as well as coregistration and can be subject to bias and over-correction.

We thank the reviewer for pointing out the potential limitations of partial volume correction. We performed visual quality on both our MRI segmentations and our MRI-PET coregistration to identify any potential issues. In addition, the analyses were performed with and without partial volume correction to ensure the robustness of our findings (see supplementary materials section 7).

3. On the other hand, the authors chose to smooth the images after partial volume correction when entering the SPM voxel-wise analysis, thereby efficiently removing the effect of the PV correction. Thus, the present results could despite partial volume correction possibly be attributed to atrophy – especially in the posterior cortical regions. Thus, the combined effect of the manipulations are obscure. Please also report data without PV correction to assess if the found results from the voxelwise analysis could result from overcorrection due to MRI segmentation flaws or if any effect was measurable.

We thank the reviewer for bringing to attention this seeming contradiction in our preprocessing methods. While it is true that broadly speaking PVC and smoothing have opposing effect of unmixing and mixing signal between voxels respectively, the post-PVC smoothing is performed on gray matter voxels only (white matter voxels are excluded from the spatially normalized image being smoothed), thus precluding gray and white matter signal from being mixed again after the smoothing. We also present results with and without PVC, and find that the observed topographies of significant deviation from controls are largely comparable with those obtained without PVC, albeit slightly more extensive spatially. We present the analyses with and without partial volume correction in supplementary materials section 7.

4. According to the method section (Subhead 2: Normative Z-scoring procedure) a regional analysis was applied without smoothing allowing for more robustly checking the results of the voxelwise analysis. However, it is unclear what the results of the regional analysis compared to the voxel-wise analysis was. Please clarify.

We thank the reviewer for pointing out the lack of clarity in our provision of the verification for the voxelwise analysis. To avoid performing an excessive number of comparisons, we attempted to validate our findings on the voxelwise group comparison against controls by performing an analogous comparison by subgroup on the principal component scores (which were derived from regional PET uptake values in subject native space without smoothing applied). The additional analysis is presented in supplementary materials section 8, table S3, and agrees with our voxelwise findings, we give greater discussion to it at the end of section 4 of results in the main manuscript.

5. Table 3 with longitudinal data is quite convincing for the reliability of the results. Please include in the header the time interval (eg. Median + range) between baseline and follow-up.

We thank the reviewer for pointing out our omission of this important information. A descriptive statistic on the time interval between baseline and follow-up visit was added to the Table 3 legend.

6. Please state clearly if any of the included subjects has been included in previous publications.

We thank the reviewer for reminding us to include this crucial detail in our manuscript. Data from subsets of both cohorts was previously analyzed and published, we added this statement to Materials & Methods, Subhead 1: Subjects.

7. The PCA found 5 significant groups. It is, however, unclear whether the deviation can be only positive or negative compared to healthy subjects or if both positive and negative deviations will be included in the same group? How does the method handle the much higher signal in centro-cingulate regions? The higher values here could skew the PCA to include these high-binding regions? What impact on the results do you think this variations in binding have?

Prior to being entered into the PCA, the regionally-averaged DVR values are actually normatively z-scored relative to controls with correction for biological (as observed among controls) effects of sex and age. Thus, variations in bindings cannot bias the PCA, because regional DVRs entering the PCA are no longer expressed in the original units, but rather in standard deviations away from controls of equivalent age and sex. The five groups of regions obtained in the PCA analysis all had only strong positive loadings because of the varimax rotation applied to the obtained components, which

simplifies the loading pattern, allowing for more straightforward definition of normative P5 and P95 cutoffs for cholinergic subgroup assignment.

8. As FDG and FEOBV has been found to correlate in AD patients (DOI: 10.1038/mp.2017.183), do you think you could have found the same results with FDG? If the found subtypes can indeed be replicated and used for future studies, the use of FDG would significantly increase the usefulness and availability of the biomarker.

We thank the reviewer for pointing out an important potential link between the patterns of changes observed in glucose metabolism among PD patients and our findings of heterogenous upregulation/downregulation in cholinergic system activity. Indeed, the findings of upregulated subcortical (especially striatal, cerebellar, and brainstem regions) glucose metabolism in the PD-related metabolic pattern (PDRP) are consistent with our observation of the most robust upregulation in cholinergic neurotransmission in those regions.

Moreover, a recent study using topographic correspondence analysis with [¹⁸F]FEOBV and [¹⁸F]FDG PET demonstrated that cortical regions exhibiting the most pronounced cholinergic basal forebrain-associated hypometabolism also show the most severe cholinergic denervation in PD. These findings provide in vivo evidence that cholinergic degeneration is a key pathological correlate of cortical hypometabolism underlying cognitive impairment in Parkinson's disease (Labrador-Espinosa et al. 2024). While it is conceivable that some of our findings could be replicated using more general neurodegeneration biomarkers, the observed upregulation of FEOBV in PD and its relationship to cerebral glucose metabolism remains speculative. Future multi-tracer studies will be necessary to clarify this association.

Reviewer #3 (Remarks to the Author):

This paper presents extensive analysis of cholinergic imaging data obtained from two large groups of PD subjects assessed at two different imaging centers. The main goal is to determine possible relations between cholinergic alterations and clinical features that could aid subtyping of PD. The results are obtained from a mixture of longitudinal and cross-sectional analysis using three different and possibly complementary approaches: (a) normative deviation system-level cholinergic covariance networks to define the subgroups, (b) voxel-level statistical parametric mapping comparisons to identify more granular regional expression patterns, and (c) whole-brain level summary statistics of cholinergic deviation spatial extent to capture the global trends of cholinergic system progression in PD.

We thank the reviewer for their thorough consideration of our work. Indeed, we believe that a multi-scale approach capturing difference both on a granular scale (voxel-level), a network scale

(principal components), and a global scale (summary measures) offers the best and most comprehensive characterization of variation in the cholinergic system among PD patients.

While the approach is comprehensive, there are concerns about potential confounds affecting the results and the overall novelty of the findings compared to what is already published

We hope that our revised manuscript (with the addition of reference region analysis as a function of a number of confounding factors) sufficiently addresses many of these concerns. We believe that the novelty of our findings lies in the comprehensive characterization of bi-directional (capturing both upregulation and downregulation) cholinergic system changes across a broad range of disease progression which we were able to examine thanks to the multi-site element of the present work.

1. Overall the paper is complex and it is hard to extract clear main messages. There is a lot of detailed information in the supplementary material, which is very hard to sift through without there being a clear summary of the relevance of each section; some of that material could be better presented in a chart form.

We thank the reviewer for constructive feedback. In the present re-submission of the manuscript, we have significantly reduced the amount of supplementary material and included infographics where appropriate to aid reader comprehension. Many of the supplementary analyses initially presented by site were omitted, because we believe that the reference region analysis presented in supplementary materials section 5 offers sufficient evidence for the validity of combining observations from both sites in a unified analysis. We still retain, however, the plot of global cholinergic system summary measures over age by subgroup which was initially tied to those analyses (see figure S11).

2. It is also hard to find clear new evidence of how cholinergic imaging contributes to subtyping over what is already known: for example, they find that the hypo-cholinergic subjects tend to be older, have poorer cognition and worse overall motor symptom severity, most of which is already known.

Indeed, there has been accumulating evidence of cholinergic system progression in PD patients. The novel contribution of the present work lies in our description of a hyper-cholinergic (upregulating) subtype in addition to the already well-known hypo-cholinergic (downregulating) subtype. Furthermore, our work is the first to perform a comprehensive multi-scale analysis of cholinergic PET data in large cohorts of patients across multiple sites, further enhancing the quality of evidence offered by the present study for the presence of cholinergic subgroups in PD.

3. An interesting observation is the finding that, as disease progresses, different regions show a different up/down-regulation trajectory, but there does not seem to be a relationship with clinical progression. Perhaps this aspect should be further expanded upon.

The results of our re-analysis, based on an updated image processing pipeline, that ensured that the contribution of confounding factors related to quantification (age and scanner influence of reference region signal) are better controlled for, actually revealed a more-or-less unified topography of longitudinal changes across the subgroups. While the hyper-cholinergic (and to a lesser extent mixed-cholinergic) subgroup exhibited evidence of upregulation relative to controls at baseline, all three subgroups appear to generally only downregulate longitudinally. The only difference that we observed was with respect to the rate of downregulation, with hyper-cholinergic subgroup demonstrating the most rapid decrease in tracer binding, followed by mixed-cholinergic, and finally hypo-cholinergic subgroup (among whom minimal longitudinal downregulation was observed, most likely due to a denervation floor effect). Our “cholinergic progression” global composite variable provides a coarse-grained unified description of the process of cholinergic changes in PD subgroups, as a loss of hyper-cholinergic and a gain of hypo-cholinergic voxels, which we used to perform exploratory correlations with clinical variables. Indeed, we found trends that broadly agree with symptoms commonly associated with cholinergic system degeneration in PD (see tables 4 & 5 in the main manuscript).

4. The merging of the two cohorts is of potential concern, even if many of them are addressed in the limitation section. The Groningen subjects are medication free at baseline, while the vast majority of the Michigan subjects are not. They have a higher UPDRS and lower Moca score in spite of a shorter disease duration (table 1). In addition, they are tested and scanned on medication at the follow-up visit, while the Michigan subjects are tested and imaged in the ‘off’ state. It is then hard to compare changes in their clinical metrics to those observed in the Michigan group. Could differences in the assessment/imaging protocol influence the findings presented in table 5?

We understand the concerns that the referee is arising from the harmonization point of view. We took this possibility very seriously, which is why a number of measures were implemented at the point of data collection to ensure that the data would be comparable between sites (such as the use of the same imaging protocol, same preprocessing pipeline being applied to both sets of images, etc). Furthermore, we performed an additional post-hoc verification that heterogeneity in reference region uptake by site, subgroup, age, visit, and scanner did not introduce a quantification bias into our

findings, and used it to inform the choice of covariates to control for in downstream analyses (discussed at length in supplementary materials section 5). From a clinical standpoint, the baseline characteristics of the cohorts differ, as they represent two distinct stages of PD: early-stage (de novo patients from Groningen) and mid-to-advanced stage (patients from the University of Michigan). This distinction is also a strength of the manuscript, as it offers a unique multicenter dataset that reflects a broad spectrum of PD patients worldwide. Despite our efforts to harmonize imaging and clinical protocols, some differences remain—for example, motor testing was performed in the ON state during follow-up in the UMCG cohort and in the OFF state at UM and hence analysis of motor MDS-UPDRS p III data was limited to the Michigan cohort as a result. These differences must be interpreted with caution, as we emphasized in the limitations section. At the same time, they reflect the real-world heterogeneity across centers, which is inherent to large-scale multicenter modeling efforts.

Further, the distribution of the data in Figs S4- S6 is quite different between the two groups (do they become more similar if the two groups are matched for disease duration?) and there did not seem to be a clear explanation for the difference.

Indeed, due to multiple substantive differences between Groningen and Michigan samples on medication status, disease duration, and motor severity, difference in distribution of global cholinergic system summary measures is to be expected. We believe these differences are related to cholinergic system progression, as evidenced by our exploratory analyses (table 4) showing for example that global loss of hyper-cholinergic voxels (particularly) and gain of hypo-cholinergic voxels was related to more advanced motor severity (as is observed in our more advance Michigan cohort), and also the primary comparison of clinical characteristics by subgroup which offered evidence of progressively greater severity of motor symptoms in mixed- and hypo-cholinergic subgroups relative to the hypo-cholinergic subgroup. We performed a post-hoc comparison evaluating the distribution of subgroup by site in the baseline cross-sectional sample (see below), where we do indeed find that the University of Groningen sample has a relative abundance of hyper-cholinergic patients (at 70.4% of total), and progressively fewer mixed-cholinergic (32%) and hypo-cholinergic (18.9%) patients. These findings are consistent with the sample differences on motor severity and the difference in distribution of global summary metrics observed in the originally included supplementary analysis by site.

Subgroup	Site		Total
	UMICH	GRON	
HYPER	21 29.6 %	50 70.4 %	71 100 %
MIXED	68 68 %	32 32 %	100 100 %

	60	14	74
HYPO	81.1 %	18.9 %	100 %
Total	149	96	245
	60.8 %	39.2 %	100 %

$\chi^2=43.993 \cdot df=2 \cdot \text{Cramer's } V=0.424 \cdot p<0.001$

If these differences in global summary measure distribution and subgroup classification were due to an effect of systematic confounds on PET signal between sites, we would expect them to manifest as a site difference in the reference region SUV comparison (presented in supplementary materials section 5), which we did not observe. Furthermore, though the analyses originally examining the distribution of global summary metrics by site and subgroup were removed from the supplementary materials section (to streamline the content, since many additional analyses had to be added to address other reviewer concerns), we repeated these analysis while including duration of disease as an additional a covariate. Duration of disease neither explained any additional variance in the distribution of global summary measures, nor did it weaken any site-specific marginal or interaction effects. We believe that is the case because self-report based measures of disease duration are an imperfect measure of true disease duration, due to unavoidable variability in clinical manifestation, variable threshold in subjective detection of earliest symptom onset across participants, and the efficacy with which participants are able to recall the time of onset. This interpretation is further corroborated by a plot of global metrics by subgroup over disease duration presented in response to a later comment, where the observed trend appears much less clear as compared to the plots of global measures by subgroup over age (as originally presented and as shown now in supplementary materials figure S11).

5. Besides possibly extending the range of disease durations, what is thus the advantage of combining the two cohorts given these potential confounds? Several comparisons between the two cohorts are given in the supplementary material, but a clear explanation for their motivation/relevance does not seem to be given. The authors present a subset of the results from analysis that was only done using the Michigan data and claim that they are overall consistent with the analysis of the entire data set. While reassuring, this still does not guarantee that combining the data is completely valid.

The purpose of combining the two cohorts is precisely to capture the broader scope (i.e., a wide range) of disease severity and progression in PD. Combining both samples allows us to offer a more comprehensive characterization of cholinergic system changes both earlier in the disease (where we suspect that cholinergic upregulation plays a key role in masking the manifestation of motor symptoms related to nigrostriatal dopaminergic losses) and later (when cholinergic downregulation may unmask symptoms). We have since taken additional precautionary measures to ensure that our samples are comparable by performing an analysis on the reference region mean

standardized uptake values (SUVs), which we would not expect to differ by site as the reference region is expected to have minimal target binding (see supplementary materials section 5). Indeed, we did not find that reference region values differed by site, which supports our analysis of the combined dataset. To further strengthen findings, we added to the supplementary materials a separate re-analysis of the voxelwise data, examining difference relative to controls and longitudinally within-subject also only in Groningen patients (see supplementary materials section 7). While the extent of upregulation does appear greater across subgroups in the Groningen cohort, the overall topography observed in the combined analysis can be recovered when the dataset from either site is considered separately, which is a very important and novel finding in this study.

6. Another concern is the very large number of comparisons – has any correction for multiple comparisons been made?

The primary aim of the present work was to define and characterize clinically and biologically the cholinergic system subgroups of PD patients. The main results of the present work are thus presented in tables 2-3 and figures 2-3. While we present uncorrected *ANOVA* *F*-test model comparison *p*-values in table 2, we added to the table legend the *Bonferroni*-adjusted α significance threshold based on the number of clinical comparisons performed between the subgroups, and adjusted our presentation of which findings in the table are significant based on this threshold with bold typeface accordingly. Table 3 is not subject to multiple comparisons correction because it tests only a single a-priori hypothesis of there being a non-random association between subgroup assignment at baseline and follow-up observation. Figures 2-3 present voxelwise statistically significant findings after voxelwise false discovery rate correction (FDR), which is a well accepted method of multiple comparison correction for mass univariate analyses in neuroimaging literature. Clinical correlation findings presented in tables 4-5 with whole brain global summary measures are largely exploratory in nature, and meant to provide a characterization of how the process of cholinergic changes on the broadest scale associates with clinical features in PD (cross-sectionally and longitudinally). For that reason, we do not provide model comparison *p*-values, and rather only report the model coefficient estimates and goodness of fit metrics (R^2) where appropriate, and bold the model coefficient where the 95% confidence interval on the estimate do not overlap with 0, to emphasize stronger trends for ease of interpretation.

7. In summary, it is believed that the manuscript would greatly benefit from streamlining, making it more readable and highlighting the main findings and their novelty. The rationale for combining the two data sets should be clearly stated.

We thank the reviewer for their many constructive suggestions, the implementation of which led to a very substantial improvement in the quality of our manuscript. We added a statement clarifying the rationale for combining the two dataset to the discussion section (see subhead 5).

Specific comments

1. Introduction

When discussing the interpretation of the tracer binding, can the author comment on different results obtained with 11C-donepezil – AchE marker (see, for example Staer et al; EJ Neurol 2024)?

AChE is expressed to a considerable extent by post-synaptic cells, so ligands targeting it (11C-donepezil) do not offer a direct measure of pre-terminals, since pathologies affecting post-synaptic cells where it is expressed may contribute to the signal. Furthermore, on account of being expressed post-synaptically it is also expressed by non-cholinergic neurons. In the striatum, for instance, AChE is expressed by dopaminergic neurons, so a component of reduced 11C-donepezil binding could be due to loss of dopaminergic innervation. These factors could blunt the chance of observing evidence for cholinergic upregulation.

2. Methods

It is stated 'The Movement Disorder Society-Revised Unified PD Rating Scale (MDS-UPDRS) motor examination was performed in the morning in the dopaminergic medication 'off' state for both cohorts at baseline, and in the 'on' state for the Groningen cohort only in the de novo subjects who received dopaminergic treatment at follow-up'. This could introduce a bias when looking at the longitudinal changes in the clinical metrics.

We thank the reviewer for their thoroughness in reading the manuscript and identifying this critical oversight in our initial submission. While we purposefully excluded site comparisons on MDS-UPDRS part III total score and derived subscores (PIGD and tremor), we erroneously still included Groningen longitudinal data for those variables in the longitudinal interval rate of change analysis now presented in table 5. This error was addressed, the Groningen data is no longer included in longitudinal analyses of clinical associations with whole brain cholinergic system progression for variables related to MDS-UPDRS part III (only Michigan data is used in those case). See also discussion above.

3. Was there any difference in the baseline data between the subjects taking different medications (Michigan)?

We performed an ANOVA *F*-test model comparison between the Michigan PD subgroups on levodopa equivalent dose at baseline to address this question. While the model comparison was statistically significant ($F(2,146)=3.545$, $p=0.03138$), the regression coefficient corresponding to MIXED vs. HYPER contrast was only marginally significant ($\beta=-0.405$ [-0.890, 0.08], $p=0.101$) and the coefficient corresponding to the HYPO vs. HYPER contrast was not statistically significant ($\beta=0.034$ [-0.459, 0.526], $p=0.893$).

4. Results

i. The Groningen group is described as having 'poorer cognition, greater impact of motor symptoms on activities of daily living, lower overall severity of motor symptoms, lower postural instability and gait difficulty (PIGD) symptom severity, lower tremor symptom severity, earlier motor disease stage, and lower disease duration from symptom onset than the Michigan cohort' While these differences are addressed in the limitation section, one still wonders if they have a significant impact on the findings.

While some of these site differences may to a limited degree be due to extraneous factors, we believe that the overall differences on disease duration, motor symptom severity (especially PIGD symptoms), and earlier motor disease stage are closely related to the underlying process of more early disease stage cholinergic changes which is primary focus of the present work.

ii. 'There was a trend for slightly lower LED among Groningen subjects relative to Michigan subjects'. This needs to be clarified: presumably it refers to the follow-up visit as these subjects are described as de-novo at baseline.

Indeed, the slight difference referred to in that section only pertains to comparisons between sites made on follow-up observations. We modified the sentence in question to make that clearer.

iii. What was the VAF for each PC?

See table below for principal component eigenvalues, difference in eigenvalue between each successive pair of components, total proportion of the variance in the data accounted for, and cumulative total variance accounted for (retained component based on cutoff of eigenvalue ≥ 1 are highlighted). Indeed, it can be observed that a relatively sharp dropoff in eigenvalue is observed after the 5th component, further justifying our

choice of eigenvalue cutoff. This can also be observed in the eigenvalue screeplot included below.

	Eigenvalue	Difference	Proportion	Cumulative
1	28.038160	20.918060	0.6095	0.6095
2	7.1201000	5.1405699	0.1548	0.7643
3	1.9795301	0.6839338	0.0430	0.8073
4	1.2955963	0.0180276	0.0282	0.8355
5	1.2775688	0.5752648	0.0278	0.8633
6	0.7023040	0.0942438	0.0153	0.8785
7	0.6080601	0.0926201	0.0132	0.8918

iv. The hyper/hypo cholinergic areas seem to vary within each subtype – can the authors comment on this? Is there a clinical correlate?

We present clinical differences between subgroups which should capture this variation in hyper- and hypo-cholinergic areas in table 2, and also present exploratory findings of correlation between the global process of loss in hyper-cholinergic and gain in hypo-cholinergic voxels in tables 4 and 5. Analysis of clinical correlates for hypo- and hyper-

cholinergic voxel topographies in specific regions is outside the scope of the present work, as its aims are to paint a broad picture of the cholinergic subgroups in PD.

v. Fig 2 – on a group level the hypo-cholinergic group shows some hypercholinergic areas – this seems mildly at odds with defining these subjects hypo-cholinergic as per authors' definition (even though this is obtained from voxel-level analysis)

Though the hypo-cholinergic subgroup does indeed retain some upregulation (particularly within the *de novo* Groningen subset within our re-analysis as presented in supplementary materials section 7), the preserved upregulation is more or less localized to the cerebellar vermis, and does not have a sufficient topographical extent to drive the systems level component above the 95th percentile of control participants. For that reason, the findings seem contradictory, but in reality they reflect (as would be expected) the need to perform analyses at multiple scales of granularity (voxel-level, systems-level, and global-level) to comprehensively characterize the more subtle deviations from controls.

vi. Fig 3 – could the comparison of the follow-up scan be made wrt to controls again as confirmation of the changes observed between baseline and follow-up? Given the relatively short time interval between baseline and follow-up, there should still be relatively good age matching between PD and HC.

In agreement with our voxel-based paired t-test analysis, the most pronounced decreases in voxel Z-scores between baseline and follow-up are observed in the hyper-cholinergic subgroup, and encompass both the weakening of upregulation (especially in the thalami, tectum, limbic structures, and the putamina) and strengthening of downregulation (most prominently observed in posterior cortices). The changes from baseline to follow-up in normative Z-scores become progressively more difficult to discern in mixed- and especially hypo-cholinergic subgroup, as consistent with less extensive topography of longitudinal within-subject changes presented in our paired-samples voxelwise t-test.

HYPHER (Baseline) vs Controls

HYPHER (Follow-up) vs Controls

HYPHER Follow-up vs. Baseline

vii. Table 4 needs to be explained better. Are the results corrected for multiple comparisons?

Table 4 is meant to give a sense of which specific directionality of cholinergic system deviation (hyper-cholinergic vs. hypo-cholinergic voxel proportion) appears to be more closely related with each given clinical outcome parameter cross-sectionally (only observations at baseline are used). Due to its exploratory nature, no model comparisons are performed and no model comparison p-values are provided, the coefficients which are bolded represent strong estimates for which the 95% interval does not overlap with 0, to aid in interpretation. For the same reason, correction for multiple comparisons is not applicable in this case, as the purpose of the table is to give a more comprehensive characterization of how global cholinergic system changes

associate with clinically relevant variables cross-sectionally, in order to inform future confirmatory investigations. We thank the reviewer for pointing out the lack of clarity in explaining what the table is meant to present to the reader, this was expanded on accordingly in the revised manuscript. A diagram was also added to the supplementary materials (figure S10) to clarify how this measure is derived from spatially normalized parametric PET images.

viii. Supplementary figures (for example S5) the hyper group has several hypo voxels (up to 20%). Can the authors comment on this? Would plotting those data as a function of disease duration provide useful information?

Indeed, the hyper group does appear to exhibit some proportion of hypo-cholinergic voxels, most likely localized to the posterior cortices as demonstrated to our voxelwise comparison against controls presented in figure 2, which are however insufficient to drive the entire principal component to fall below the 5th percentile of controls. We plotted the global cholinergic system measures over disease duration (see below), but find that it paints a less straightforward picture. This may be so because determination of disease duration depends on patient self-report of symptom onset, which is an imperfect measure of duration. Age is also imperfect due to variability in onset, but it does not depend on self-report, which is possibly why the plot of these global measures over age (as provided in figure S11 now) seems to paint a clearer picture.

ix. Table 5 – see comment on Delta-UPDRS above. Are any of the results adjusted for multiple comparisons? Which regressions are overall significant? Some R2 values are very low.

No model comparisons were performed due to the exploratory nature of longitudinal rate of change analyses presented in table 5, and for that same reason multiple comparison correction is not applicable in the present context, and no claims can be made about which models are overall significant (no specific hypothesis test was performed). Of greatest interest may be the models where the coefficient estimate are strong (95% confidence intervals on the estimate don't overlap with 0) for the association between rate of change in global cholinergic system progression (ΔC) and rate of change in the clinical outcome parameter (ΔY), which would be MoCA scores, executive function Z-score, gait velocity, and turning time (as measured via inertial sensors). Those are the variables where more prominent longitudinal loss of hyper-cholinergic and gain of hypo-cholinergic voxels leads to more prominent longitudinal deteriorations in the clinical parameters (faster decrease in MoCA, executive function, gait speed and faster increase in turning time from baseline to follow-up visit).

Low R^2 values on the cognitive outcome parameters (MoCA scores and executive Z-scores) is likely reflective of their multi-factorial nature, i.e. other neuromodulatory systems and mechanisms which we are not capturing may be independent major contributors. Decreases in gait speed and difficulties with turning are more specific symptoms of cholinergic deficits in more advanced PD patients (especially those with episodic gait disturbances like freezing of gait) which we reported on in prior [^{11}C]PMP and [^{18}F]FEOBV correlation studies (Bohnen et al. 2013; Landau 2014; Mancini et al. 2017) which is why a higher rate of global progression in cholinergic brain changes explains a greater proportion of variance in longitudinal rate of change of those parameters specifically (~30% of variance in rate of longitudinal clinical parameter change).

References

- Bohnen, Nicolaas I., Kirk A. Frey, Stephanie Studenski, Vikas Kotagal, Robert A. Koeppe, Peter J. H. Scott, Roger L. Albin, and Martijn L. T. M. Müller. 2013. "Gait Speed in Parkinson Disease Correlates with Cholinergic Degeneration." *Neurology* 81 (18): 1611–16.
- Bohnen, Nicolaas I., Prabesh Kanel, Robert A. Koeppe, Carlos A. Sanchez-Catasus, Kirk A. Frey, Peter Scott, Gregory M. Constantine, Roger L. Albin, and Martijn L. T. M. Müller. 2021. "Regional Cerebral Cholinergic Nerve Terminal Integrity and Cardinal Motor Features in Parkinson's Disease." *Brain Communications* 3 (2): fcab109.
- Bohnen, Nicolaas I., Stiven Roytman, Prabesh Kanel, Martijn L. T. M. Müller, Peter J. H. Scott, Kirk A. Frey, Roger L. Albin, and Robert A. Koeppe. 2022. "Progression of Regional Cortical Cholinergic Denervation in Parkinson's Disease." *Brain Communications* 4 (6): fcac320.

- Labrador-Espinosa, Miguel A., Jesús Silva-Rodríguez, Niels Okkels, Laura Muñoz-Delgado, Jacob Horsager, Sandra Castro-Labrador, Pablo Franco-Rosado, et al. 2024. "Cortical Hypometabolism in Parkinson's Disease Is Linked to Cholinergic Basal Forebrain Atrophy." *Molecular Psychiatry*, December. <https://doi.org/10.1038/s41380-024-02842-9>.
- Landau, William M. 2014. "Gait Speed in Parkinson Disease Correlates with Cholinergic Degeneration." *Neurology*. Ovid Technologies (Wolters Kluwer Health).
- Mancini, Martina, Katrijn Smulders, Rajal G. Cohen, Fay B. Horak, Nir Giladi, and John G. Nutt. 2017. "The Clinical Significance of Freezing While Turning in Parkinson's Disease." *Neuroscience* 343 (February): 222–28.